# Communication-Efficient FL via Model-Agnostic Projection Optimization

## Abstract

Federated learning (FL) enables collaborative model training across distributed clients without sharing sensitive data. However, communication overhead remains a significant bottleneck, particularly for large-scale models. Low-rank decomposition techniques address this by approximating each layer's weights or gradients with a product of low-rank matrices, thereby reducing the communication cost in FL. While effective, these methods are constrained by the layer's architecture and shapes, limiting their flexibility and performance. We propose *Model-Agnostic Projection Optimization* (MAPO), a novel method that reshapes and factorizes the full model gradient into a *fixed reconstruction matrix* and a *trainable projection vector*, avoiding layer-wise decomposition and architecture constraints. MAPO directly optimizes the projection in a randomly sampled subspace, with all clients generating the reconstruction matrix via a shared random seed, incurring no additional communication overhead for synchronization. By decoupling the gradient from architectural constraints through reshaping and enabling communication-free exploration of dynamic subspaces via seed sharing, MAPO provides a more flexible and efficient low-rank representation. Empirical results demonstrate the effectiveness of MAPO in various FL settings.

## 1 Introduction

Federated Learning (FL) is a distributed framework that enables model training across many clients without centralizing data. In each communication round, clients download a global model, update it using local data, and send modifications back to the server, which aggregates them (e.g., via FedAvg (McMahan et al., 2017)). While this iterative process enables collaborative learning, frequent transmission of model updates incurs significant communication overhead, limiting FL applications, particularly with large models or resource-constrained clients.

Communication-Efficient Federated Learning (CEFL) literature (Jia et al., 2025) proposes a vast range of strategies to reduce communication load. Konečný (2016) categorizes them into *sketched updates*, which compress the total model update after optimization (e.g., subsampling, quantization, random projection), and *structured updates*, which restrict the trainable parameters to a lower-dimensional subspace before optimization (e.g., random masks, weight-sharing, and low-rank decomposition).

**Low-rank decomposition** is a widely used approximation technique that expresses model gradients or parameters as the product of low-rank matrices (Sainath et al., 2013). *Parameter decomposition* is particularly effective for Parameter-Efficient Fine-Tuning (PEFT), where auxiliary low-rank adaptation modules (LoRA) are added to each layer to reduce the computation and storage overhead of full-model fine-tuning (Hu et al., 2021). Although LoRA alleviates communication burdens in FL, constraining model parameters to a low-rank subspace can degrade performance. In contrast, *gradient decomposition* preserves full-rank model representations during inference and restricts only the gradients to a low-rank form during backpropagation (Wang et al., 2018b; Jaderberg et al., 2014; Lebedev et al., 2014; Denil et al., 2013). A visual comparison is shown in Figure 1.

**Challenges.** While CEFL methods for gradient decomposition (Vogels et al., 2019; Zhao et al., 2023b; Park & Klabjan, 2024; Guo et al., 2024a; Hu et al., 2024), parameter decomposition (Yao et al., 2021; Hyeon-Woo et al., 2021; Jeong & Hwang, 2022; Hameed et al., 2023; Zhao et al., 2023a), or LoRA variants (Sun et al., 2024; Zhang et al., 2023; Zhu et al., 2024; Hao et al., 2024; Guo et al., 2024b) offer notable benefits, they face several key challenges:

Figure 1: Comparison of various decomposition methods, from left: no decomposition, low-rank parameter decomposition, frozen model with low-rank adapter (LoRA), low-rank gradient decomposition, and MAPO.

1) The layer-wise decomposition that adheres to the structural constraints (e.g., fully connected or convolutional), requiring *architecture-dependent* implementation for each layer decomposition. 2) Given a decomposition $\Delta W_i \in \mathbb{R}^{d_1 \times d_2} \approx B_i A_i$, where $A_i \in \mathbb{R}^{r \times d_2}$ and $B_i \in \mathbb{R}^{d_1 \times r}$, the number of transmitted parameters is $\mathbf{C} = |A_i| + |B_i| = r(d_1 + d_2)$ for $r \in \mathbb{N}$, restricting the communication rate to multiples of $(d_1 + d_2)$, imposing a *rigid communication granularity* as $C \in (d_1 + d_2)\mathbb{N}$. 3) Given $M$ number of clients and $(A_i^j, B_i^j)$ denoting the low-rank decomposition of layer $i$ from client $j$, averaging these low-rank matrices is *not equivalent to full-rank aggregation* as:

$$\frac{1}{M}(B_i^1 A_i^1 + B_i^2 A_i^2 + \cdots + B_i^M A_i^M) \neq \frac{1}{M}(B_i^1 + B_i^2 + \cdots + B_i^M)\frac{1}{M}(A_i^1 + A_i^2 + \cdots + A_i^M).$$

This mismatch arises precisely when clients maintain different reconstruction matrices (i.e. $A_i^j \neq A_i^k$). 4) Although fixing all $\{A_i^j\}_{j=1}^M$ matrices to the same values can mitigate the aggregation problem and improve the communication granularity to $\mathbf{C} \in d_1\mathbb{N}$, as shown in FA-LoRA (Sun et al., 2024) and EvoFed (Rahimi et al., 2024), it restricts the model's ability to explore richer subspaces, often leading to *suboptimal solutions* (Guo et al., 2024b). Thus, we aim to answer the following key question:

*How can we develop an architecture-independent model-wide decomposition that offers flexibility on communication rate, address the low-rank averaging problem, and suboptimality of freezing $A$?*

**Key Ideas.** We propose a novel Model-Agnostic Projection Optimization (**MAPO**) that streamlines gradient projection and addresses its challenges while being computationally lighter than layer-wise methods. Our key ideas are described as follows:

**(i)** Firstly, MAPO reimagines low-rank gradient projection by treating the entire model gradient as a single matrix rather than layer-by-layer decomposition. It eliminates architecture-specific constraints by merging the flattened gradients of all layers, constructing the *universal gradient vector* $\Delta W \in \mathbb{R}^d$.

**(ii)** Secondly, given any communication budget $k$, MAPO pads $\Delta W$ with zeros so the length becomes divisible by $k$. Afterwards, padded $\Delta W$ will be reshaped to $\Delta W' \in \mathbb{R}^{k \times \lceil d/k \rceil}$ which further can be decomposed it into a $A \in \mathbb{R}^{1 \times \lceil d/k \rceil}$ and $B \in \mathbb{R}^{k \times 1}$ matrices, as $\Delta W' = BA$.

**(iii)** Lastly, instead of freezing $A$, MAPO refreshes $A$ each round using a shared random seed. This ensures that all clients use the same $A$ during aggregation, avoiding low-rank averaging error, while still allowing exploration of richer subspaces across rounds without additional communication.

**Summary of Contributions.** By integrating **(i)** model-level decomposition, **(ii)** flexible communication rate, and **(iii)** subspace exploration, MAPO offers a flexible trade-off between communication cost and performance while remaining more efficient than low-rank decomposition methods. Figure 3 illustrates the distinction between MAPO and other paradigms. Our main contributions are:

- We introduce model-agnostic optimization of gradient projections that enhances communication and computation efficiency, boosts performance through exploration, and offers more flexibility in balancing communication and error rate.

- We provide a theoretical analysis of MAPO convergence behavior and establish its computational efficiency compared to layer-wise factorization with the same communication and error rates.

- We conduct extensive experiments across diverse datasets, model architectures, and baselines, demonstrating that MAPO surpasses existing methods in full training and fine-tuning scenarios.

## 2 BACKGROUND AND RELATED WORKS

In this section, we review key CEFL approaches in relation to MAPO. We begin with sketched update techniques that project model updates into subspaces, outlining their limitations. Then, we examine structured update methods, particularly projection optimization, highlighting the unique opportunities and challenges introduced by operating within a fixed subspace.

## 2.1 SKETCHED UPDATE VS. STRUCTURED UPDATE

**Sketched update** includes techniques such as sparsification (Konečný, 2016), quantization (Bernstein et al., 2018; Lin et al., 2018; Reisizadeh et al., 2020; Sun et al., 2020), gradient subspace projection (Azam et al., 2021; Oh et al., 2022; Park & Choi, 2023), and random subspace projection (Shi & Eryilmaz, 2021; Rahimi et al., 2024). They aim to compress the information in the update vector $\Delta W \in \mathbb{R}^d$ defined as the difference between the locally optimized and the global model $\Delta W = W^* - W_g$.

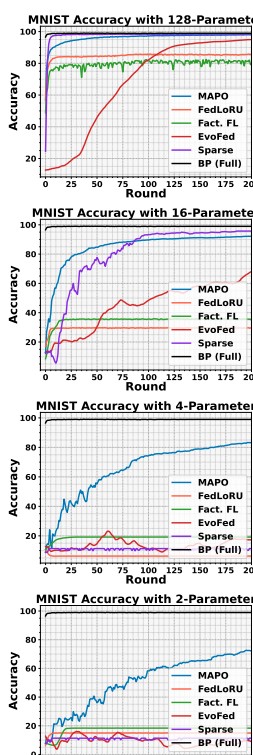

The subspace projection process (Shi & Eryilmaz, 2021; Woodruff, 2014; Li et al., 2018) defines a random matrix $A \in \mathbb{R}^{p \times d}$, and finds the projection vector $B \in \mathbb{R}^p$, which minimizes the reconstruction error $\|\Delta W - BA\|_2$, where $d$ denotes the total number of model parameters and $p \ll d$ is compressed length:

$$B^* = \arg\min_{B \in \mathbb{R}^p} \|\Delta W - BA\|_2 \quad ; \quad B^* \approx \Delta W \mathbf{A}^\top (\mathbf{A}\mathbf{A}^\top)^{-1}.$$

As the matrix $A$ is considerably large ($p \times d$), various methods propose novel designs for $A$ to adapt it for large-scale models. Notably, defining $A$ as a subset of seen gradient vectors results in a significantly lower rank of $A$ that suffices for an effective projection (Azam et al., 2021; Park & Choi, 2023). More recently, EvoFed (Rahimi et al., 2024) utilizes evolutionary strategies to evolve $A$, improving its representation capacity.

**Sketching Limitations.** Although sketched methods benefit from a full-rank training, their shortcoming is blindness to the loss surface $\mathcal{L}(W; \mathcal{D})$ and alternative solutions besides $\Delta W$ that can be reconstructed from the projection subspace. They typically perform well, given a sufficiently large subspace, but as the compression rate increases, the projection vector reconstruction ends up far off from $\Delta W$. In contrast, subspace

Figure 2: MNIST performance for varying number of trainable parameters.

optimization directly finds the steepest direction within the subspace, leading to a more effective loss reduction. Figure 2 illustrates a centralized MNIST training, showing the performance degradation of sketched techniques, such as EvoFed and sparsification, compared to MAPO. As sparsity increases, MAPO continues to converge, even having 2 or 4 trainable parameters out of 11,274.

**Structured update** techniques reduce the number of trainable parameters and communication cost by constraining the weights or gradients to a low-rank subspace by structural modification such as pruning (Han et al., 2015; He et al., 2017; Luo et al., 2017; Zhang et al., 2018), weight–sharing (Chen et al., 2015; Courbariaux et al., 2016; Ullrich et al., 2017), low-rank gradient (Vogels et al., 2019; Zhao et al., 2023b; Park & Klabjan, 2024; Guo et al., 2024a; Hu et al., 2024), and parameter decomposition (Yao et al., 2021; Hyeon-Woo et al., 2021; Jeong & Hwang, 2022; Hameed et al., 2023; Zhao et al., 2023a), including LoRA and its variants (Hu et al., 2021; Zhang et al., 2023; Sun et al., 2024; Zhu et al., 2024; Hao et al., 2024). Although parameter decomposition techniques reduce the model size and representation, resulting in subpar performance for general training, as shown in Figure 2 for Factorized-FL (Jeong & Hwang, 2022). Therefore, CEFL generally adopts a gradient decomposition direction. In particular, gradient decomposition methods with a fixed $A$, also known as *projection optimization* (Denil et al., 2013; Jaderberg et al., 2014; Lebedev et al., 2014; Wang et al., 2018b).

Prior works on gradient decomposition relied on each layer's shape and architecture, producing a unique $A_i$ and $B_i$ matrices for each layer, limiting the feasibility of sharing a projection matrix $A$ across layers. MAPO overcomes this limitation by evenly partitioning the whole model gradient vector $\Delta W \in \mathbb{R}^d$ into $k$ segments $\{\Delta W_i'\}_{i=1}^k \in \mathbb{R}^{k \times \lceil d/k \rceil}$, allowing the use of a shared random reconstruction matrix $A \in \mathbb{R}^{1 \times \lceil d/k \rceil}$ across all partitions, maintaining the benefits of model-wide projection while substantially reducing memory costs.

## 2.2 PARAMETER-EFFICIENCY VS. COMMUNICATION-EFFICIENCY

Despite their apparent similarities, parameter decomposition and gradient decomposition methods differ fundamentally in assumptions and objectives. Parameter decomposition directly imposes a low-rank structure on the model parameters, effectively replacing the original model with a compressed version. Although this reduces the total number of parameters and computation, it still requires transmitting all parameters at each communication round, resulting in no relative reduction in

Figure 3: Step-by-Step illustration of methodology based on propositions, demonstrating how each step will contribute to designing MAPO factorization and differing from LoRA architecture.

communication per parameter. In contrast, gradient decomposition methods maintain the original model architecture and computational complexity but substantially reduce communication overhead by transmitting compressed updates that are significantly smaller than the whole model.

In this work, to ensure a fair assessment of communication efficiency, we evaluate MAPO against gradient-based compression baselines under consistent model architectures. Additional experiments with parameter decomposition and LoRA-based methods are provided in Appendices B and C for completeness. Key methodological distinctions among related works are summarized in Table 1.

Table 1: Summary of CEFL methods and objectives. The column "Comm. Flex" indicates support for arbitrary bitrates, and "Agg. Eq." denotes equivalence between low-rank and full-rank averaging.

| Method | Scope | Target | Full-rank Inference | Agg. Eq. | PEFT | Fixed Subspace | Arch-Agnostic | Comm Flex | Personalized FL |
|---|---|---|---|---|---|---|---|---|---|
| Sparsification (Konečný, 2016) | Model | Update | ✓ | ✓ | ✗ | ✗ | ✓ | ✓ | ✗ |
| Quantization (Reisizadeh et al., 2020) | Model | Update | ✓ | ✓ | ✗ | ✗ | ✓ | ✓ | ✗ |
| EvoFed (Rahimi et al., 2024) | Model | Update | ✓ | ✓ | ✗ | ✓ | ✓ | ✓ | ✗ |
| Factorized-FL (Jeong & Hwang, 2022) | Layer | Parameter | ✗ | ✗ | ✗ | ✗ | ✗ | ✗ | ✓ |
| LoRA (Hu et al., 2021) | Layer | Adapter | ✗ | ✗ | ✓ | ✗ | ✗ | ✗ | ✗ |
| FA-LoRA (Sun et al., 2024) | Layer | Adapter | ✗ | ✓ | ✓ | ✓ | ✗ | ✗ | ✗ |
| SA-LoRA (Guo et al., 2024b) | Layer | Adapter | ✗ | ✗ | ✓ | ✗ | ✗ | ✗ | ✓ |
| FedLoRU (Park & Klabjan, 2024) | Layer | Gradient | ✓ | ✓ | ✗ | ✓ | ✗ | ✗ | ✗ |
| **MAPO (Ours)** | Model | Gradient | ✓ | ✓ | ✗ | ✓ | ✓ | ✓ | ✗ |

## 3 PROPOSED METHOD

In this section, we introduce MAPO and its application in FL. We first present the MAPO factorization technique and discuss its key properties regarding communication efficiency and error rate. Subsequently, we detail how MAPO can be effectively integrated into the FL training process.

### 3.1 MODEL-AGNOSTIC PROJECTION OPTIMIZATION (MAPO)

**MAPO Description.** MAPO provides model-agnostic factorization of the global model gradient $\Delta W \in \mathbb{R}^d$, avoiding architecture-specific constraints and enabling subspace exploration during optimization. As illustrated in Figure 1, MAPO reshapes the universal gradient $\Delta W \in \mathbb{R}^{d \times 1}$ into $\Delta W' \in \mathbb{R}^{k \times \lceil d/k \rceil}$, which is then decomposed into a reconstruction vector $A \in \mathbb{R}^{1 \times \lceil d/k \rceil}$ and a projection vector $B \in \mathbb{R}^{k \times 1}$. It is equivalent to partitioning $\Delta W$ into $k$ segments and sharing a fixed reconstruction matrix $A$ across all partitions. This design preserves model-wide projection benefits while substantially reducing memory overhead. Figure 3 shows a step-by-step visualization analogous to Propositions 3.4 to 3.6.

**MAPO Properties.** MAPO aims to construct an expressive subspace, enabling a small $B$ to encode sufficient information for updating the model efficiently. First, we formally define the concepts of communication overhead rate and reconstruction error rate in the context of matrix factorization in Definitions 3.2 and 3.3. Using these definitions, Proposition 3.4 establishes that reshaping a single layer preserves both the factorization error and communication rates. Extending this, Proposition 3.5 demonstrates that vectorizing multiple layers into a single matrix similarly maintains these properties. Finally, this leads to the proof of Proposition 3.6, which introduces a computationally and communication-efficient, model-agnostic factorization method as an alternative to traditional layer-wise gradient projection techniques. Appendix I presents the formal proofs.

**Lemma 3.1 (Gaussian Matrices are Full Rank).** *Let $A \in \mathbb{R}^{m \times n}$ be a random matrix with entries drawn independently from a Gaussian distribution $\mathcal{N}(0, \sigma^2)$. Then, $A$ is almost surely of full rank, i.e., $\text{rank}(A) = \min(m, n)$, as the probability of $A$ being rank deficient is zero. This result follows from standard properties of random matrices (Vershynin, 2018; Tao, 2012).*

Figure 4: Application of MAPO to communication-efficient FL.

**Definition 3.2** (**Communication Overhead Rate**). *Let $\Delta W_i \in \mathbb{R}^{d_1 \times d_2}$ be the update matrix of a model. Suppose the factorization of $\Delta W_i$ as $\Delta W_i = B_i A_i$, where $A_i \in \mathbb{R}^{q \times d_2}$ is a fixed random matrix and $B_i \in \mathbb{R}^{d_1 \times q}$ is a trainable matrix with $q \leq \min(d_1, d_2)$ being the factorization rank. The* **communication overhead rate** $\mathrm{CO}_{rate}$ *is defined as the ratio of the size of $B_i$ to the size of $\Delta W$:*

$$\mathrm{CO}_{rate} = \frac{\mathrm{size}(B_i)}{\mathrm{size}(\Delta W_i)} = \frac{q}{d_2}.$$

**Definition 3.3** (**Reconstruction Error Rate**). *Using the same factorization as Definition 3.2, the reconstruction error rate is the expected ratio of the reconstruction error to the original model update. Given full-rank random reconstruction (Lemma 3.1), it is expressed as:*

$$\frac{\mathbb{E}_{A_i}\left[\|\Delta W_i - B_i A_i\|_2^2\right]}{\|\Delta W_i\|_2^2} = 1 - \frac{q}{d_2}.$$

**Proposition 3.4** (**Single-Vector Factorization**). *Let $\Delta W_i$, $A_i$, and $B_i$ be factorizations of a single layer of the network as in Definition 3.2. By reshaping $\Delta W_i$ into $\Delta W_i' \in \mathbb{R}^{1 \times d_1 d_2}$ the factorization of $\Delta W_i' = B_i' A_i'$ where $A_i' \in \mathbb{R}^{p \times d_1 d_2}$ and $B_i' \in \mathbb{R}^{1 \times p}$ can achieve the same* **reconstruction error** *and* **communication overhead** *to the conventional factorization of $\Delta W_i$ when $p = q d_1$.*

**Proposition 3.5** (**Multi-Layer Factorization**). *Let $\Delta W_i$, $A_i$, and $B_i$ be* **single-vector factorization** *of $i$-th layer of the $N$-layered network as in Proposition 3.4. By concatenating the reshaped weights $\Delta W_i$ into $\Delta W' \in \mathbb{R}^{1 \times d}$, where $d = \sum_{i=1}^{N} d_1^i d_2^i$. The factorization of $\Delta W' = B' A'$ where $A' \in \mathbb{R}^{p \times d}$ and $B' \in \mathbb{R}^{1 \times p}$ can achieve the same* **reconstruction error** *and* **communication overhead** *to the single-vector factorization applied to each $\Delta W_i$ when $p = N q d_1$.*

**Proposition 3.6** (**MAPO Factorization**). *Let $\Delta W$, $A$, $B$, and rank $p$ be a multi-layer factorization of a network as defined in Proposition 3.5. By reshaping $\Delta W \in \mathbb{R}^{1 \times d}$ into $\Delta W' \in \mathbb{R}^{k \times \lceil d/k \rceil}$, and the factorization of $\Delta W' = B' A'$ where $A' \in \mathbb{R}^{1 \times \lceil d/k \rceil}$ and $B' \in \mathbb{R}^{k \times 1}$, we can achieve the same* **reconstruction error** *and* **communication overhead** *to the multi-layer factorization of $\Delta W$ when $k = p$, while reducing the size of reconstruction matrix by a factor of $k^2$.*

### 3.2 APPLICATION TO COMMUNICATION-EFFICIENT FEDERATED LEARNING

This subsection explains how our method, outlined in Section 3.1, is utilized in FL. The procedure pseudo-code is provided in Algorithm 1, and visualized in Figure 4.

**Matrix Construction and Broadcasting.** To ensure consistency across the network, the server and all clients start from an identical condition at each round. We guarantee identical model parameters $W_t$ and reconstruction matrix $A_t$ by broadcasting a random seed $r_t$ and the aggregated projection vector $\overline{B}_t$ at the beginning of round $t$. The initial aggregated projection vector is set to $\overline{B}_0 = \mathbf{0}$.

**In the first round** ($t = 0$), all clients and the server initialize the model $W^0$ using the same seed. The reconstruction matrix $A^0 \in \mathbb{R}^{1 \times \lceil d/k \rceil}$ is drawn from Gaussian $A^0 \sim \mathcal{N}(0, I)$, and the client $j$'s projection vector $B^{0,j} \in \mathbb{R}^{k \times 1}$ is set to 0 for all $1 \leq j \leq M$, where $M$ is the total number of clients.

**In subsequent rounds** ($t \geq 1$), clients update their local model $W^t$ using the previous round's matrix $A^{t-1}$, the model parameters $W^{t-1}$, and the broadcasted projection vector $\overline{B}^t$ as follows:

$$W^t = W^{t-1} + \mathbf{vec}(\overline{B}^t A^{t-1})_{[0:d]}, \tag{1}$$

where $\mathbf{vec}(\cdot)$ and $(\cdot)_{[0:d]}$ denotes vectorization and truncating to the first $d$ elements. Clients then regenerate $A^t \sim \mathcal{N}(0, I)$ using the seed $r^t$ and reset $B^{t,j} \leftarrow \mathbf{0}$, ensuring $A^t$ and $W^t$ synchronization.

---

**Algorithm 1:** Federated Learning with MAPO

---

**Input** : Initial random seed $r^0$, global model $W^0$, reconstruction matrix $A^0$, projection vector $\overline{B}^0$
**Output** : Final global model $W^T$

1 Initialize all clients and server with the same seed $r^0$;
2 Initialize $W^0 \in \mathbb{R}^d$, $A^0 \in \mathbb{R}^{1 \times \lceil d/k \rceil}$, $\overline{B}^0 \leftarrow \mathbf{0} \in \mathbb{R}^{k \times 1}$;
3 **for** each communication round $t = 1, \ldots, T - 1$ **do**
4    **Server:** Broadcast $\overline{B}^{t-1}$ and seed $r^{t-1}$ to all clients;
5    **for** each **Client** $j = 1, \ldots, M$ (in parallel) **do**
6      Receive $\overline{B}^{t-1}$ and $r^{t-1}$;
7      Update local model: $W^t \leftarrow W^{t-1} + \mathbf{vec}(\overline{B}^t A^{t-1})[0:d]$;
8      Re-generate $A^t = \mathcal{N}(0, \sigma^2 I_d)\big|r^{t-1}$;
9      Initialize $B^{t,j} \leftarrow \mathbf{0} \in \mathbb{R}^{k \times 1}$;
10      **for** each local epoch $e = 1, \ldots, E$ **do**
11        Compute gradient: $\nabla B^{t,j} \leftarrow \nabla_{B^{t,j}} \mathcal{L}^j(W^t + \mathbf{vec}(B^{t,j} A^{t-1})[0:d], \mathcal{D}^j)$;
12        Update projection vector: $\hat{B}^{t,j} \leftarrow B^{t,j} - \eta \nabla B^{t,j}$;
13        Set $B^{t,j} \leftarrow \hat{B}^{t,j}$;
14      **end**
15      Send $\hat{B}^{t,j}$ to the server;
16    **end**
17    **Server:**
18    Re-generate $A^t = \mathcal{N}(0, \sigma^2 I_d)\big|r^{t-1}$;
19    Aggregate: $\overline{B}^t \leftarrow \frac{1}{S} \sum_{j=1}^M b_j \hat{B}^{t,j}$, where $S = \sum_j b_j$;
20    Update global model: $W^{t+1} \leftarrow W^t + \mathbf{vec}(\overline{B}^t A^{t-1})[0:d]$;
21    Generate new seed $r^t$ (e.g., $r^t = \text{hash}(r^{t-1})$);
22 **end**
23 **return** $W^T$;

---

**Local Projection Optimization.** This step optimizes the projection $\hat{B}^{t,j}$ to minimizes the client loss $\mathcal{L}(W^t + \mathbf{vec}(B^{t,j} A^{t-1})_{[0:d]}, \mathcal{D}^j)$, where $\mathcal{D}^j$ denotes client $j$'s local dataset, and model weights are derived as $W^t + \mathbf{vec}(B^{t,j} A^t)_{[0:d]}$ given the random matrix $A^t$.

At each communication round $t \geq 1$, after initializing $A_t$ and $B^{t,j}$, clients perform local training to optimize $B^{t,j}$ using their local data $\mathcal{D}^j$. The gradient of the projection vector is computed as:

$$\nabla B^{t,j} = \nabla_{B^{t,j}} \mathcal{L}^j(W^t + \mathbf{vec}(B^{t,j} A^{t-1})_{[0:d]}) \quad \text{for} \quad \mathcal{L}^j(W) = \frac{1}{|\mathcal{D}^j|} \sum_{x \in \mathcal{D}^j} \ell(W, x). \quad (2)$$

where $\ell(W, x)$ is the loss function (e.g., cross-entropy loss) given model $W$ and data point $x$. Therefore, given the learning rate $\eta$, only the projection $\hat{B}^{t,j}$ is updated using gradient descent as:

$$\hat{B}^{t,j} \leftarrow B^{t,j} - \eta \nabla B^{t,j}, \quad (3)$$

After optimization, clients send their optimized projection vector $\hat{B}^{t,j}$ to the server. The low dimensionality of $\hat{B}^{t,j}$ compared to $W^t$ results in communication efficiency.

**Server-Side Aggregation and Global Model Update.** Upon receiving the projection vectors $\hat{B}^{t,j}$ and their corresponding weights $b^j = |D^j|$ (e.g., batch sizes or number of local samples) from the clients, the server aggregates them to form the global projection vector:

$$\overline{B}^t = \frac{1}{S} \sum_{j=1}^M b^j \hat{B}^{t,j}, \quad \text{for} \quad S = \sum_{j=1}^M b_j \quad (4)$$

This weighted averaging captures the collective contribution of all clients, proportional to their data sizes. The server then broadcasts the aggregated projection vector $\overline{B}^t$ to all clients. After receiving $\overline{B}^t$, the server and all clients update their local models using the reconstruction matrix $A^t$ and the aggregated projection vector $\overline{B}^t$ as:

$$W^{t+1} = W^t + \mathbf{vec}(\overline{B}^t A^{t-1})_{[0:d]}. \quad (5)$$

This update integrates the clients' optimized directions into their local models and ensures synchronization across the network. This process is repeated until the global model converges.

Table 2: Summary of datasets and models used in our experiments.

| Dataset | Client Distribution | Train/Test | # Classes | Model | # Parameters |
|---|---|---|---|---|---|
| MNIST (LeCun et al., 1998) | Non-IID (2 classes) | 60K / 10K | 10 | CNN - 2 Layers | 11,274 |
| FMNIST (Xiao et al., 2017) | Non-IID (2 classes) | 60K / 10K | 10 | CNN - 2 Layers | 11,274 |
| CIFAR-10 (Krizhevsky et al., 2009) | Non-IID (2 classes) | 50K / 10K | 10 | CNN - 4 Layers | 1,146,634 |
| CIFAR-100 (Krizhevsky et al., 2009) | Non-IID (10 classes) | 50K / 10K | 100 | WideResNet 16d4w | 2,854,420 |
| TinyImageNet (University, 2015) | Non-IID (10 classes) | 100K / 10K | 200 | WideResNet 16d4w | 2,880,120 |
| Shakespeare (Caldas et al., 2018) | Distributed by Roles | 14K / 2K | 65 | LSTM | 814,957 |
| Sentiment140 (Caldas et al., 2018) | Distributed by Users | 1.4M / 200K | 2 | Transformer | 2,221,570 |
| GLUE Tasks (Wang et al., 2018a) | Non-IID | *differ per task* | *differ per task* | RoBERTa-Large | 357,199,876 |

## 4 CONVERGENCE ANALYSIS

We analyze the convergence behavior of FL with MAPO.

**Assumption 4.1.** *For each $j$, $\mathcal{L}^j(v)$ is $\beta$-smooth, i.e., $\left\|\nabla\mathcal{L}^j(u) - \nabla\mathcal{L}^j(v)\right\| \leq \beta\|u-v\|$ for any $u, v$.*

**Assumption 4.2.** *Variance of the stochastic gradient of $D^j$ is bounded for each client $j$, i.e.,*

$$\mathbb{E}\left[\left\|\nabla\mathcal{L}^j(W) - \widetilde{\nabla}\mathcal{L}^j(W)\right\|^2\right] \leq \sigma_l^2$$

**Assumption 4.3.** *Bounded clients' gradient dissimilarity: $\frac{1}{M}\sum_{j=1}^M \left\|\nabla\mathcal{L}^j(W) - \nabla\mathcal{L}(W)\right\|^2 \leq \sigma_g^2$.*

**Assumption 4.4.** *At each communication round $t$, the server selects a subset $S_t \subset [M]$ with $|S_t| = m < M$ clients uniformly and the sampling variance is $\sigma_{het}^2 = \rho\sigma_g^2$ where $\rho = \frac{M-m}{m(M-1)}$.*

**Theorem 4.5.** *Let Assumptions 4.1 to 4.4 hold, and suppose $\eta_t \leq \frac{1-4\epsilon}{4\beta(1+\epsilon)}$. Then, after $T$ communication rounds each with $E$ local steps, the following bound holds:*

$$\frac{1}{4H_T}\sum_{t=0}^{T-1}\eta_t\,\mathbb{E}\left[\left\|\nabla\mathcal{L}(W^t)\right\|^2\right] \leq \frac{\mathbb{E}[\mathcal{L}(W^0)] - \mathcal{L}^*}{EH_T} + 2E(\epsilon + \beta + \beta\epsilon)\left(\sigma_l^2 + \sigma_g^2 + \sigma_{het}^2\right)\frac{1}{H_T}\sum_{t=0}^{T-1}\eta_t^2,$$

*where $H_T = \sum_{t=0}^{T-1}\eta_t$, $\epsilon$ is the JL distortion parameter, and $\mathcal{L}^*$ is the minimum of $\mathcal{L}(W)$.*

With a decreasing learning rate satisfying $\sum_{t=0}^{\infty}\eta_t \to \infty$, $\sum_{t=0}^{\infty}\eta_t^2 < \infty$ (e.g., $\eta_t = \frac{\eta_0}{t+c}$ for some constants $\eta_0 > 0$, $c > 0$), the term $H_T = \sum_{t=0}^{T-1}\eta_t$ grows unbounded, while the weighted sum $\sum_{t=0}^{T-1}\eta_t^2$ remains finite. Therefore, the right-hand side of Theorem 4.5's bound satisfies

$$\frac{\mathbb{E}[\mathcal{L}(W^0)] - \mathcal{L}^*}{H_T} \to 0, \quad \frac{1}{H_T}\sum_{t=0}^{T-1}\eta_t^2 \to 0 \quad \text{as } T \to \infty,$$

confirming convergence to a stationary point, as the gradient norm average satisfies

$$\frac{1}{H_T}\sum_{t=0}^{T-1}\eta_t\,\mathbb{E}\left[\|\nabla\mathcal{L}(W^t)\|^2\right] \to 0.$$

As shown above, the convergence bound is influenced by the factor $\epsilon + \beta + \beta\epsilon$, and becomes tightest when $\epsilon = 0$, i.e., in the absence of reconstruction error. The proof is located in Appendix J.

## 5 EXPERIMENTAL SETUP

We evaluate MAPO across diverse model architectures, tasks, and baselines. The benchmarks span five image classification datasets, MNIST (LeCun et al., 1998), FMNIST (Xiao et al., 2017), CIFAR-10, CIFAR-100 (Krizhevsky et al., 2009), and TinyImageNet (University, 2015), as well as sequential tasks, including next-character prediction on Shakespeare and sentiment analysis on Sentiment140, both drawn from the LEAF benchmark suite (Caldas et al., 2018), tailored for FL. Additionally, we evaluate MAPO as a fine-tuning method, alongside LoRA baselines on various GLUE (Wang et al., 2018a) tasks, highlighting the communication and computation efficiency in Appendix B. The dataset specifications and corresponding model architectures are summarized in Table 2, highlighting MAPO's adaptability across varying data modalities, model scales, and application domains.

**Non-IID Distribution.** To simulate realistic FL conditions, we partition the training datasets in a non-IID manner across 100 clients. For image classification and GLUE tasks, each client is assigned a distinct subset of classes. For LEAF tasks, we follow the natural user-based partitioning, where individual Shakespearean roles and Twitter users correspond to separate clients.

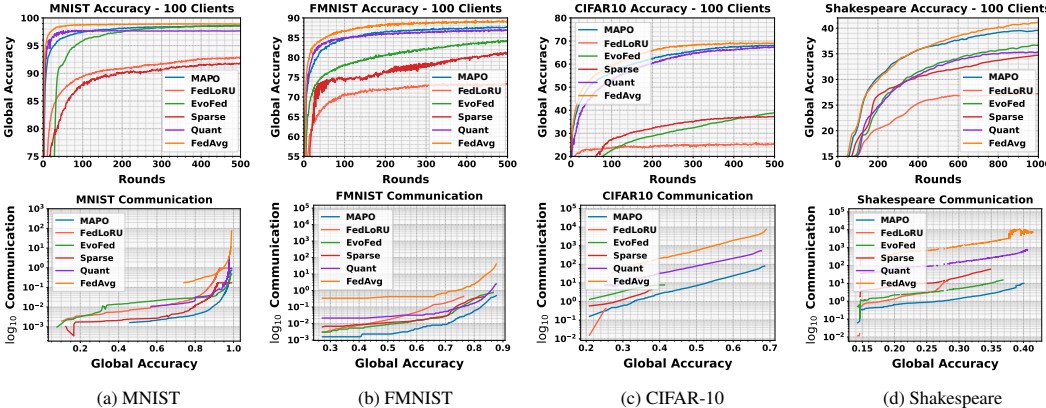

Figure 5: **Performance comparison** of all methods on MNIST, FMNIST, CIFAR-10, and Shakespeare datasets. The top row shows the accuracy, while the bottom row illustrates the communication cost per accuracy.

Table 3: Summary of maximum accuracy (%) and communication cost (% relative to FedAvg). Accuracy values report mean (±std) over 3 runs, estimated from observed variance.

| Method | MNIST Com. | MNIST Acc. | FMNIST Com. | FMNIST Acc. | CIFAR-10 Com. | CIFAR-10 Acc. | CIFAR-100 Com. | CIFAR-100 Acc. | Shakespeare Com. | Shakespeare Acc. | Sent140 Com. | Sent140 Acc. | TinyImageNet Com. | TinyImageNet Acc. |
|---|---|---|---|---|---|---|---|---|---|---|---|---|---|---|
| FedAvg | 100 | 98.9 (±0.1) | 100 | 89.2 (±0.2) | 100 | 69.0 (±0.2) | 100 | 43.47 (±0.3) | 100 | 41.86 (±0.3) | 100 | 74.90 (±0.3) | 100 | 36.48 (±0.4) |
| Sparse | 15.3 | 92.1 (±0.4) | 24.1 | 81.1 (±0.4) | 2.7 | 37.15 (±0.5) | 1.20 | 33.72 (±0.5) | 1.73 | 34.86 (±0.4) | 1.93 | 74.21 (±0.3) | 1.32 | 25.34 (±0.5) |
| Quantize | 31.3 | 97.6 (±0.2) | 24.1 | 87.1 (±0.3) | 15.2 | 67.40 (±0.3) | 6.10 | 40.05 (±0.4) | 10.11 | 35.45 (±0.4) | 13.85 | 73.70 (±0.3) | 8.75 | 34.47 (±0.4) |
| EvoFed | 9.40 | 98.5 (±0.2) | 7.60 | 84.7 (±0.3) | 3.4 | 39.50 (±0.4) | 20.4 | 37.62 (±0.4) | 0.23 | 36.76 (±0.3) | 0.40 | 70.50 (±0.5) | 1.85 | 15.40 (±0.5) |
| FedLoRU | 30.2 | 93.8 (±0.4) | 17.9 | 74.1 (±0.5) | 1.7 | 23.52 (±0.5) | 1.20 | 19.10 (±0.5) | 1.67 | 28.07 (±0.5) | 1.30 | 66.61 (±0.4) | 1.27 | 7.31 (±0.5) |
| **MAPO** | **2.95** | **98.6** (±0.1) | **3.10** | **88.0** (±0.2) | **1.20** | **68.3** (±0.2) | **0.91** | **40.16** (±0.3) | **0.13** | **39.96** (±0.3) | **0.19** | **74.50** (±0.2) | **0.97** | **35.22** (±0.3) |

**Model Architectures.** We evaluate MAPO across diverse architectures of varying complexity, including CNNs for MNIST, FMNIST, and CIFAR-10, WideResNet for CIFAR-100 and TinyImageNet, LSTM for next-character prediction, Transformer for sentiment analysis, and RoBERTa for GLUE tasks. Detailed architecture specifications and hyperparameters are in Appendix D.

**Baselines.** We compare MAPO against multiple baselines, including standard compression methods with subsampling (Sparse) (Konečný, 2016) and quantization (Quant) (Reisizadeh et al., 2020), EvoFed (Rahimi et al., 2024), and FedLoRU (Park & Klabjan, 2024). Subsampling and quantization serve as references to establish MAPO's performance compared to conventional compression techniques. EvoFed provides a strong comparison to demonstrate the effectiveness of MAPO's subspace optimization relative to methods applying compression post-optimization. FedLoRU allows us to highlight MAPO's dynamic subspace exploration and its benefits over static layer-wise gradient projections. Results comparing MAPO with additional parameter-factorization (Factorized-FL (Jeong & Hwang, 2022)) and adapter-based fine-tuning baselines (LoRA (Hu et al., 2021), FA-LoRA (Sun et al., 2024), and SA-LoRA (Guo et al., 2024b)) are included in Appendices B and C.

**Federated Learning Setting.** In each training round, 10% of the clients are randomly selected to participate. Selected clients train locally in parallel and transmit their updates to the central server, which aggregates these updates and redistributes the resulting global model back to the clients.

## 6 RESULTS AND DISCUSSIONS

We now discuss our experimental results in detail and provide insights into MAPO's performance. Figure 5 (top row) shows the accuracy of MAPO compared to multiple baseline methods across various datasets. MAPO consistently outperforms all other methods and achieves accuracy comparable to FedAvg, despite transmitting only a fraction of the parameters. This improvement results from MAPO's dynamic subspace optimization, which promotes effective exploration and efficient use of the communication budget to minimize the loss function directly. Additionally, Figure 5 (bottom row) illustrates the minimal communication cost required by each method to reach a given accuracy level, highlighting MAPO's significantly lower communication demands (logarithmic scale on the y-axis). Additional results on CIFAR-100, TinyImageNet, and Sentiment140 are presented in Appendix A.

Table 3 summarizes experimental results by comparing the maximum accuracy of each baseline and their communication cost relative to FedAvg. To ensure fair comparison, communication costs are reported as the percentage required to reach the accuracy of the worst-performing baseline. MAPO consistently achieves competitive accuracy with significantly lower communication overhead. Specifically, on MNIST and FMNIST, MAPO achieves 99.6% and 98.6% of FedAvg's accuracy, respectively, using only 3% of FedAvg's communication cost. For CIFAR-10, CIFAR-100, and

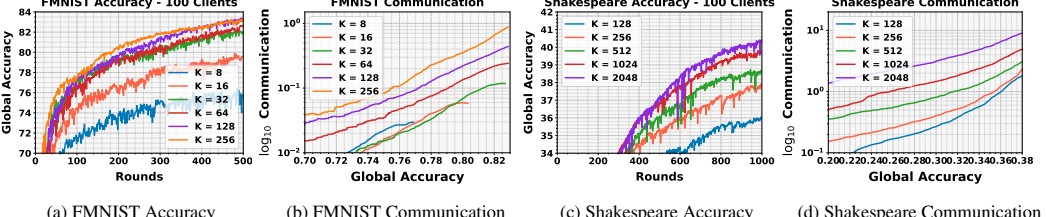

(a) FMNIST Accuracy     (b) FMNIST Communication     (c) Shakespeare Accuracy     (d) Shakespeare Communication

Figure 6: Accuracy and communication cost per accuracy level for FMNIST and Shakespeare datasets. Demonstrating the effect of a number of trainable parameters ($k$) on the communication efficiency of MAPO.

TinyImageNet, MAPO attains 98.9%, 92.4%, and 96.5% of FedAvg accuracy, respectively, while consuming approximately 1% of the communication. Finally, in sequential tasks (Shakespeare and Sentiment140), MAPO retains up to 95.5% and 99.5% of FedAvg's accuracy, respectively, while dramatically reducing communication to less than 0.2%.

**MAPO Hyperparameter.** MAPO simplifies gradient projection by applying a single factorization across all model parameters, thus replacing per-layer rank with a single hyperparameter, $k$, directly controlling communication cost and model accuracy. Figure 6 illustrates the effect of $k$ on performance and communication efficiency for the FMNIST and Shakespeare datasets. While a smaller $k$ significantly reduces communication overhead, it slows the convergence, requiring more training rounds. Conversely, increasing $k$ improves convergence speed and accuracy but rapidly raises communication costs, often with diminishing returns. Therefore, the optimal $k$ achieves a target accuracy with minimal total communication. Figure 6(b) and (c) show communication costs associated with specific accuracy levels, guiding the selection of optimal $k$. We use the same guidelines for all baselines to fairly tune hyperparameters. (See Appendix D.1)

**Fresh Reconstruction Matrix.** A key factor in MAPO's performance is using a dynamically generated reconstruction matrix $A$ rather than a fixed one. This approach promotes the exploration of new subspaces throughout training. Figure 7 illustrates the benefits of using a fresh $A$ on the FMNIST and Shakespeare datasets. We evaluate MAPO across varying numbers of trainable parameters, ranging from $2^0$ to $2^{13}$. For FMNIST, this corresponds to $0.009\%$ to $72.27\%$ of the total model parameters, while for Shakespeare, it spans from $0.0001\%$ to nearly $1\%$. In both cases, MAPO with a fresh $A$ achieves superior convergence with fewer parameters, effectively leveraging the search space. In contrast, when $A$ is frozen, performance follows a logarithmic correlation with the number of trainable parameters, requiring an exponentially larger parameter count to match the results obtained with a fresh $A$.

**Additional Results.** Comparisons with LoRA-based methods and Factorized-FL are provided in Appendices B and C. Appendix E supplements our main experiments with evaluations under IID distributions and without client sampling. The ablation study on the effect of input dimension and model parameters on communication rate and training stability during exploration of $A$ is presented in Appendices F and G. Finally, Appendix K presents a detailed analysis of memory complexity, emphasizing computational efficiency and flexibility compared to layer-wise low-rank factorization.

(A) FMNIST

(B) Shakespeare

Figure 7: Comparison of having a fresh vs. frozen $A$.

**Limitations.** MAPO's improved communication efficiency comes with additional computational overhead from gradient projection optimization. While significantly reduced compared to prior methods, MAPO still requires $\lceil d/r \rceil + r$ memory and computation (instead of $dr + r$; see Appendix K). MAPO complements, but does not replace, PEFT methods like LoRA, as it reduces communication overhead without decreasing the trainable parameters or storage requirements (see Appendix B).

## 7 CONCLUSION

We introduced *Model-Agnostic Projection Optimization* (MAPO), a novel approach for CEFL. Unlike layer-wise decomposition, MAPO factorizes the entire gradient using a projection vector and a random reconstruction matrix, regenerated at each round. MAPO balances communication efficiency and accuracy without imposing architecture-specific constraints or fixed-subspace limitations. Our theoretical analysis establishes convergence guarantees, and empirical results demonstrate superior performance and scalability across diverse datasets, confirming its practical value for FL.

## REPRODUCIBILITY STATEMENT

We provide complete, anonymous source code and configuration files in the supplementary materials: (i) a PyTorch project for LoRA and GLUE fine-tuning experiments, and (ii) a JAX project for all other federated experiments. The algorithmic design of MAPO is specified in Section 3.1 with a step-by-step visualization (Figures 1 and 3) and a federated pseudo-code listing in Algorithm 1 (see also the "Matrix Construction and Broadcasting" paragraph in Section 3.2 for seed handling and synchronization). Dataset/model choices and train/test splits appear in Table 2, with the non-IID partitioning and training protocol detailed in Section 5 ("Non-IID Distribution" and "Model Architectures"), and hyperparameters in Appendix D. Theoretical assumptions and guarantees are stated in Assumptions 4.1 to 4.4 and Theorem 4.5, with complete proofs in Appendices I and J. The supplementary packages include all configurations and environmental specifications necessary to reproduce all reported results within the stated variance.

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

# A    ACCURACY AND COMMUNICATION LEARNING CURVES

This appendix provides extended experimental results that complement the main findings discussed in Section 5. We include detailed evaluations of MAPO and baseline methods on CIFAR-100, TinyImageNet, and Sentiment140 datasets. Similar to the main results, Figure 8 reports both maximum test accuracy and the communication cost required to reach a given accuracy threshold. These additional experiments further demonstrate MAPO's superior communication efficiency and consistent performance gains across more challenging and large-scale tasks.

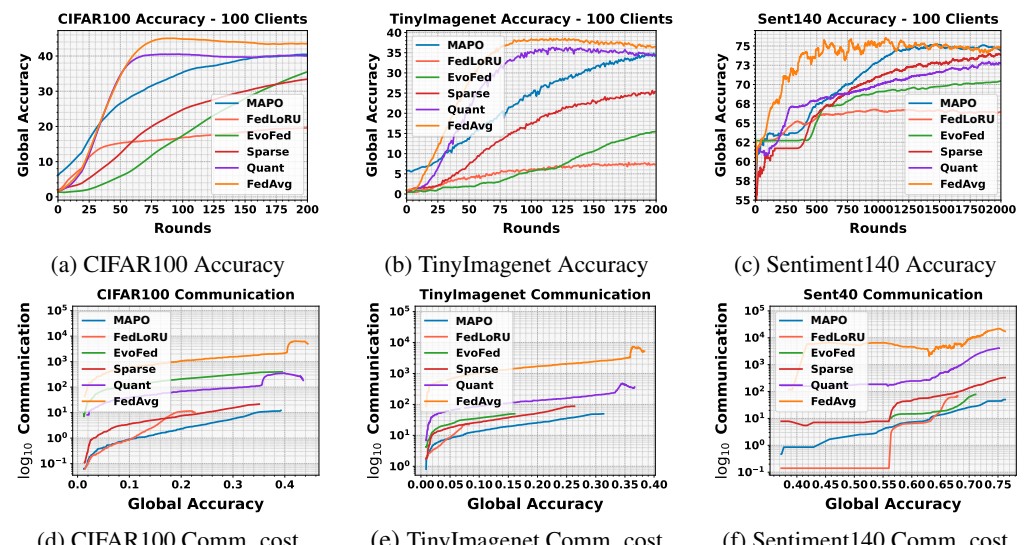

(a) CIFAR100 Accuracy       (b) TinyImagenet Accuracy       (c) Sentiment140 Accuracy

(d) CIFAR100 Comm. cost     (e) TinyImagenet Comm. cost     (f) Sentiment140 Comm. cost

Figure 8: **Performance comparison** of MAPO and baseline methods on CIFAR100, TinyImagenet, and Sentiment140 datasets. The top row shows the accuracy achieved by each method on the respective datasets, while the bottom row illustrates the communication cost associated with each method.

# B    COMPARISON WITH LOW-RANK ADAPTATION IN FINE-TUNING

We conduct fine-tuning experiments using RoBERTa-large on five GLUE tasks to evaluate MAPO alongside LoRA, FA-LoRA, and SA-LoRA. Table 4 compares the number of trainable parameters and the communication load per round for each method. Table 5 summarizes fine-tuning results under federated settings, reporting communication efficiency based on the number of rounds and total communication required to reach 80% accuracy. Overall, the results indicate that MAPO improves communication efficiency without compromising performance.

Table 4: Number of trainable and communication parameters per round for different methods.

| Method | Number of trainable parameters | Number of communication parameters per round |
|---|---|---|
| LoRA | 1.83M | 0.78M |
| FA-LoRA | 1.44M | 0.39M |
| SA-LoRA | 1.83M | 0.39M |
| MAPO$_{d/1k}$ | 357M | 0.36M |
| MAPO$_{d/10k}$ | 357M | 35.70K |
| MAPO$_{d/100k}$ | 357M | 3.57K |
| MAPO$_{d/1m}$ | 357M | 357 |

Table 5: Comparison of model accuracies, communication rounds, and total communication cost.

| Model | SST2 | | | QNLI | | | RTE | | | MNLIm | | | MNLImm | | |
|---|---|---|---|---|---|---|---|---|---|---|---|---|---|---|---|
| | Acc | Round | Total | Acc | Round | Total | Acc | Round | Total | Acc | Round | Total | Acc | Round | Total |
| LoRA | 84.86 | 36 | 28.08M | 91.72 | 85 | 66.30M | 86.62 | 180 | 140.40M | 87.41 | 86 | 67.08M | 87.34 | 82 | 63.96M |
| FA-LoRA | 94.15 | 44 | 17.16M | 91.63 | 76 | 29.64M | 57.28 | — | — | 85.92 | 76 | 29.64M | 86.46 | 213 | 83.07M |
| SA-LoRA | 95.41 | 19 | 7.41M | 91.04 | 55 | 21.45M | 70.01 | — | — | 89.44 | 29 | 11.31M | 85.49 | 126 | 49.14M |
| MAPO$_{d/1k}$ | 96.79 | 5 | 1.78M | 93.14 | 11 | 3.93M | 87.91 | 23 | 8.21M | 88.90 | 17 | 6.07M | 88.26 | 22 | 7.85M |
| MAPO$_{d/10k}$ | 96.10 | 5 | 178.50K | 92.57 | 8 | 285.60K | 89.57 | 23 | 821.10K | 88.81 | 18 | 642.60K | 87.43 | 25 | 892.50K |
| MAPO$_{d/100k}$ | 95.53 | 5 | 17.85K | 89.24 | 7 | 24.99K | 84.38 | 24 | 85.68K | 85.04 | 20 | 71.40K | 84.60 | 29 | 103.53K |
| MAPO$_{d/1m}$ | 90.37 | 7 | 2.50K | 80.09 | 34 | 12.14K | 57.04 | — | — | 72.46 | — | — | 37.76 | — | — |

## C  COMPARISON WITH FACTORIZED-FL

In this section, we present a detailed comparison between MAPO and Factorized-FL as a representative of the parameter decomposition methods. Factorized-FL can be interpreted as a variant of rank-1 LoRA, where a sparse bias matrix substitutes for LoRA's frozen fine-tuned weights, initialized to zero. Table 6 reports the communication efficiency of MAPO and Factorized-FL on CIFAR-10 and SVHN datasets, evaluated under both IID and non-IID partitions. Each column denotes the total communication in GB required to reach X% of FedAvg's final test accuracy. Results show that MAPO achieves significantly lower communication costs compared to Factorized-FL while maintaining competitive performance across both datasets and data distributions.

Table 6: Communication cost comparison across different methods on SVHN and CIFAR-10 under IID and Non-IID settings.

| Method | SVHN | | | | CIFAR-10 | | | | Com/Round |
|---|---|---|---|---|---|---|---|---|---|
| | IID@80% | IID@90% | Non-IID@80% | Non-IID@90% | IID@80% | IID@90% | Non-IID@80% | Non-IID@90% | |
| FedAvg | 183.51 | 244.68 | 285.46 | 509.75 | 305.85 | 407.80 | 326.24 | 652.48 | 20.39GB |
| Factorized-FL | 127.75 | 182.50 | 146.00 | 219.00 | 182.50 | 292.00 | 200.75 | 310.25 | 18.25GB |
| MAPO$_{2k}$ | 0.32 | 0.79 | 0.56 | – | 0.32 | – | 0.94 | – | **0.78MB** |
| MAPO$_{16k}$ | **0.08** | **0.18** | **0.12** | **0.27** | **0.08** | **0.18** | **0.23** | **0.45** | 6.25MB |
| MAPO$_{40k}$ | 3.84 | 8.64 | 5.76 | 13.12 | 3.84 | 8.64 | 10.88 | 21.12 | 0.32GB |

## D  IMPLEMENTATION DETAILS AND HYPERPARAMETERS

All experiments were conducted on a single NVIDIA RTX 3090 with 24 GB of memory. The main experiments and baselines are implemented with JAX (Bradbury et al., 2018). The GLUE tasks and LLM fine-tuning implementation use Hugging Face libraries and models implemented in FederatedScope (Xie et al., 2023) with half precision (i.e., 16-bit float). The model configuration and training used in this work are provided in Tables 7 and 8.

Table 7: Neural network configurations for different datasets.

| Dataset | Model type | # Conv | Kernel | Hidden features | # Linear | # Output | # Parameters |
|---|---|---|---|---|---|---|---|
| MNIST | CNN | 2 | 5×5 | 8, 16 | 1 | 10 | 11.3K |
| FMNIST | CNN | 2 | 5×5 | 8, 16 | 1 | 10 | 11.3K |
| CIFAR-10 | CNN | 4 | 5×5 | 64, 64, 128, 128 | 2 | 10 | 1.1M |
| CIFAR-100 | WideResNet | 16 | 3×3 | 64×4, 128×4 | 2 | 100 | 2.8M |
| TinyImageNet | WideResNet | 16 | 3×3 | 64×4, 128×4 | 2 | 200 | 2.88M |
| Shakespeare | LSTM | - | - | 256, 8 (embed) | 2 | 65 | 814K |
| Sentiment140 | Transformer | - | - | 512, 96 (embed) | 2 | 2 | 2.2M |
| SVHN | CNN | 4 | 5×5 | 64, 64, 128, 128 | 2 | 10 | 1.1M |
| GLUE | RoBERTa-large | - | - | 1024 (hidden) | 2 | Varies | 357M |

Table 8: Training hyperparameters for FedAvg and variants.

| Hyperparameter | MNIST | FMNIST | CIFAR-10 | CIFAR-100 | TinyImageNet | Sentiment140 | Shakespeare | SVHN | GLUE |
|---|---|---|---|---|---|---|---|---|---|
| Batch size | 32 | 32 | 32 | 32 | 32 | 32 | 32 | 32 | 128 |
| Optimizer | SGD | SGD | SGD | AdamW | AdamW | SGD | SGD | SGD | SGD |
| Learning rate | 0.2 | 0.2 | 0.03 | 0.1 | 0.2 | 0.001 | 0.2 | 0.03 | 0.02 |
| Momentum | 0.9 | 0.9 | 0.4 | 0.9 | 0.9 | 0.9 | 0.9 | 0.4 | 0.0 |
| L1 regularization | 0.0 | 0.0 | 1e-4 | 0.0 | 1e-5 | 0.0 | 5e-6 | 1e-4 | 0.0 |
| L2 regularization | 0.0 | 0.0 | 1e-5 | 3e-3 | 1e-4 | 0.0 | 5e-5 | 1e-5 | 0.0 |

### D.1  GUIDELINE FOR TUNING THE MAPO $k$

To select an optimal value of $k$, we begin by evaluating MAPO under a high compression setting, typically starting around $k \approx \sqrt{d}$. We then increase $k$ progressively by doubling its value at each step. After each evaluation, we compare both the final accuracy and the convergence rate. The search continues until we observe that increases in $k$ no longer produce meaningful gains in performance and the faster convergence no longer compensates for the higher communication cost. In practice, this means choosing the smallest $k$ that achieves near optimal accuracy while also minimizing the total number of communication rounds. For example, if $k = 128$ reaches 90 percent accuracy in 100 rounds and $k = 256$ reaches the same accuracy in 60 rounds, the choice of $k = 128$ remains preferable because the overall communication load is lower despite the slower convergence.

# E    IID AND CLIENT SAMPLING

This section includes the results of additional experiments on IID distribution and client sampling for MNIST, FMNIST, and CIFAR-10. Across all three datasets, we observe consistent trends. Reducing the fraction of clients participating (from all clients to 10%) moderately decreases accuracy for all methods, and non-IID settings introduce additional accuracy penalties. However, MAPO's performance remains robust in these more demanding scenarios; it routinely stays close to FedAvg's high-accuracy results while maintaining significant communication savings. This resilience suggests that MAPO's approach scales well to heterogeneous data distributions and partial-participation regimes, crucial in large-scale FL deployments.

Table 9: Extrapolated MNIST results for IID vs. non-IID and full vs. 10% client participation.

| | IID | | | | Non-IID | | | |
| | All clients | | 10% clients | | All clients | | 10% clients | |
| Method | Com. | Acc. | Com. | Acc. | Com. | Acc. | Com. | Acc. |
|---|---|---|---|---|---|---|---|---|
| FedAvg | 100% | 99.6% | 100% | 99.5% | 100% | 99.3% | 100% | 98.9% |
| Sparse | 10.0% | 93.9% | 12.0% | 93.6% | 13.3% | 93.4% | 15.3% | 92.1% |
| Quantize | 22.0% | 98.8% | 25.0% | 98.5% | 29.0% | 98.2% | 31.3% | 97.6% |
| EvoFed | 6.5% | 99.4% | 7.0% | 99.2% | 8.5% | **99.0%** | 9.4% | **98.5%** |
| FedLoRU | 22.0% | 95.0% | 25.0% | 94.7% | 28.2% | 94.3% | 30.2% | 93.8% |
| MAPO | **2.0%** | **99.5%** | **2.3%** | **99.3%** | **2.7%** | **99.0%** | **2.9%** | **98.5%** |

Table 10: Extrapolated FMNIST results for IID vs. non-IID and full vs. 10% client participation.

| | IID | | | | Non-IID | | | |
| | All clients | | 10% clients | | All clients | | 10% clients | |
| Method | Com. | Acc. | Com. | Acc. | Com. | Acc. | Com. | Acc. |
|---|---|---|---|---|---|---|---|---|
| FedAvg | 100% | 91.5% | 100% | 91.0% | 100% | 90.0% | 100% | 89.2% |
| Sparse | 16.0% | 84.0% | 19.0% | 83.5% | 21.0% | 82.0% | 24.1% | 81.1% |
| Quantize | 16.0% | 89.7% | 19.0% | 89.2% | 21.0% | 88.0% | 24.1% | 87.1% |
| EvoFed | 4.5% | 87.0% | 5.5% | 86.5% | 6.8% | 85.5% | 7.6% | 84.7% |
| FedLoRU | 12.0% | 76.8% | 14.0% | 76.2% | 15.5% | 75.0% | 17.9% | 74.1% |
| MAPO | **2.0%** | **90.0%** | **2.3%** | **89.6%** | **2.7%** | **88.8%** | **3.1%** | **88.0%** |

Table 11: Extrapolated CIFAR-10 results for IID vs. non-IID and full vs. 10% client participation.

| | IID | | | | Non-IID | | | |
| | All clients | | 10% clients | | All clients | | 10% clients | |
| Method | Com. | Acc. | Com. | Acc. | Com. | Acc. | Com. | Acc. |
|---|---|---|---|---|---|---|---|---|
| FedAvg | 100% | 73.0% | 100% | 72.0% | 100% | 70.0% | 100% | 69.0% |
| Sparse | 1.8% | 41.0% | 2.0% | 40.0% | 2.4% | 38.0% | 2.7% | 37.2% |
| Quantize | 10.0% | 71.0% | 12.0% | 70.0% | 13.0% | 68.5% | 15.2% | 67.4% |
| EvoFed | 2.0% | 43.0% | 2.5% | 42.0% | 3.0% | 40.5% | 3.4% | 39.5% |
| FedLoRU | 1.1% | 27.0% | 1.3% | 26.0% | 1.5% | 24.5% | 1.7% | 23.5% |
| MAPO | **0.8%** | **71.5%** | **0.9%** | **70.8%** | **1.0%** | **69.2%** | **1.2%** | **68.3%** |

# F   ABLATION STUDY: INPUT DIMENSION AND COMPRESSION RATE

This section reports additional experiments designed to quantify how the projection dimension $k$ should scale as the task becomes more complex (higher model capacity and higher input dimensionality). We evaluate on the HAM10000 skin cancer detection dataset with both low- and high-resolution inputs and two representative architectures.

## EXPERIMENTAL SETUP

We consider four model–input configurations and assign $k$ commensurate with their complexity:

$$\textbf{CNN (3M params)}, 28 \times 28 : \ k = 2^{12}, \qquad \textbf{CNN (208M params)}, 224 \times 224 : \ k = 2^{18},$$

$$\textbf{WRN (2.8M params)}, 28 \times 28 : \ k = 2^{12}, \qquad \textbf{WRN (2.8M params)}, 224 \times 224 : \ k = 2^{16}.$$

We compare FedAvg (uncompressed) to representative communication-efficient baselines (Sparse, Quant, EvoFed, FedLoRU) and **MAPO**. We report (i) test accuracy (**Acc.**, %) and (ii) normalized communication cost (**Com.**, %), where $\underline{\text{Com.} = 100}$ is the per-round uplink payload of FedAvg for the corresponding model–resolution configuration (lower is better).

| | **CNN (3M)**, $28 \times 28$, $k = 2^{12}$ | | **CNN (208M)**, $224 \times 224$, $k = 2^{18}$ | | **WRN (2.8M)**, $28 \times 28$, $k = 2^{12}$ | | **WRN (2.8M)**, $224 \times 224$, $k = 2^{16}$ | |
| Method | Acc. | Com. | Acc. | Com. | Acc. | Com. | Acc. | Com. |
|---|---|---|---|---|---|---|---|---|
| FedAvg | 77.05 | 100.00 | 79.76 | 100.00 | 79.13 | 100.00 | 81.83 | 100.00 |
| Sparse | 71.47 | 1.67 | 74.09 | 2.07 | 73.57 | 1.72 | 79.07 | 13.72 |
| Quant | 75.51 | 10.31 | 78.23 | 14.21 | 77.82 | 5.63 | 79.32 | 28.44 |
| EvoFed | 71.82 | 4.57 | 74.13 | 3.83 | 73.97 | 10.81 | 78.11 | 18.72 |
| FedLoRU | 74.60 | 1.33 | 78.18 | 1.51 | 77.01 | 1.52 | 78.98 | 12.43 |
| **MAPO** | **76.58** | **1.07** | **79.20** | **1.23** | **78.20** | **0.93** | **80.16** | **9.27** |

Table 12: **HAM10000: accuracy–communication trade-offs across model/input complexity and projection $k$.** Each block fixes a model and input resolution; **Com.** is per-round uplink normalized to the corresponding FedAvg (100). As complexity rises (from 28×28 to 224×224 and/or higher-capacity models), larger $k$ is used to maintain fidelity. **MAPO** consistently matches or exceeds the accuracy of other compression baselines while operating at substantially lower communication budgets.

**Scaling trend.** Higher input resolution and model capacity necessitate a larger projection dimension $k$ to preserve gradient information under projection. This is reflected in the chosen $k$ across the four settings.

**Robust trade-off.** Even as $k$ increases for the more complex settings, **MAPO** attains a favorable accuracy–communication balance. For example:

CNN (208M), $224 \times 224$: MAPO reaches 78.20% at only 0.93% communication, vs. FedAvg 79.76% at 100%.

WRN (2.8M), $224 \times 224$: MAPO achieves 80.16% at 9.27% communication, vs. FedAvg 81.83% at 100%.

**Consistency across regimes.** On low-resolution tasks (e.g., $28 \times 28$), MAPO preserves accuracy close to FedAvg while reducing communication by two orders of magnitude; on high-resolution tasks, it remains competitive and generally superior to other compression baselines.

## G  ABLATION STUDY: SUBSPACE EXPLORATION AND STABILITY

**Design.** We compare three variants of MAPO based on how the reconstruction vector $A$ is handled:
(1) FRESH-A, which regenerates a new random $A_t$ every communication round;
(2) FROZEN-A, which keeps $A$ fixed throughout training; and
(3) TRAINABLE-A, which optimizes $A$ via gradient descent.

**Subspace exploration.** In theory, refreshing $A_t$ expands the number of explored low-rank subspaces, which can accelerate optimization under a tight communication budget. Empirically, we observe the same behavior across all six benchmarks: FRESH-A may begin slightly slower but consistently surpasses FROZEN-A after the initial rounds and converges to substantially higher accuracy. This validates that exploration dominates early-round variance.

**Trainable-A: communication limitations and instability.** Optimizing $A$ (Trainable-A) introduces a fundamental limitation: It removes the possibility of communication-less synchronization of the reconstruction vector. Unlike FRESH-A and FROZEN-A, which synchronize $A$ using shared random seeds, TRAINABLE-A requires every client to communicate both a projected update and the trainable vector, leading to the following communication rate:

$$\text{COM}_{\text{Trainable-A}} = \frac{d}{k'} + k',$$

which is minimized when $k' = \sqrt{d}$, yielding a lower bound of $2\sqrt{d}$ per round.

In large models (e.g., GLUE with $d = 357M$ parameters), this corresponds to $2\sqrt{d} \approx 37{,}789$ scalars per round, far exceeding the communication cost of MAPO with $k \in \{d/10{,}000,\ d/100{,}000\} = \{35{,}700,\ 3{,}570\}$, both of which outperform other baselines. A similar gap appears in lightweight vision and text tasks. For instance:

$$\text{COM}_{\text{FMNIST}}^{\text{Trainable-A}} = 213, \quad \text{COM}_{\text{Shakespeare}}^{\text{Trainable-A}} = 1806,$$

while MAPO requires less to follow the FedAvg performance.

$$\text{COM}_{\text{FMNIST}} = 32, \quad \text{COM}_{\text{Shakespeare}} = 256.$$

Although this additional communication could, in principle, improve convergence, our experiments show the opposite. Trainable-A often converges faster locally but fails to converge globally due to the **low-rank averaging problem**: clients' optimized $A_t$ vectors diverge in incompatible directions, and the global aggregation collapses into unstable or oscillatory updates. Even when stabilizing aggregation using full-rank averaging at the server and transmitting the full model back to clients, the final accuracy remains far below the FRESH-A variant and is typically close to FROZEN-A, despite requiring much higher communication, as shown in Figure 9.

**Interpretation.** We hypothesize that regenerating $A_t$ promotes more robust exploration of diverse low-rank subspaces, especially under tight communication budgets, than gradient-based optimization of $A$. While this behavior makes MAPO effective, it also highlights a limitation of parameter-factorization methods such as LoRA, where the underlying model parameters are fixed and do not permit $A$-regeneration. The broader implications for both gradient and parameter factorization remain open for future investigation.

A fixed small subspace is quickly capacity-limited. Allowing $A$ to refresh each round enhances subspace exploration, resulting in reliably higher accuracies after the initial transient. In our implementation, $A$ is (re)initialized once per communication round. Each round comprises many local optimization steps (multiple batches/epochs), executed within a fixed subspace. The subspace changes only after server aggregation. This design avoids the instability that could arise from re-drawing $A$ at every local step.

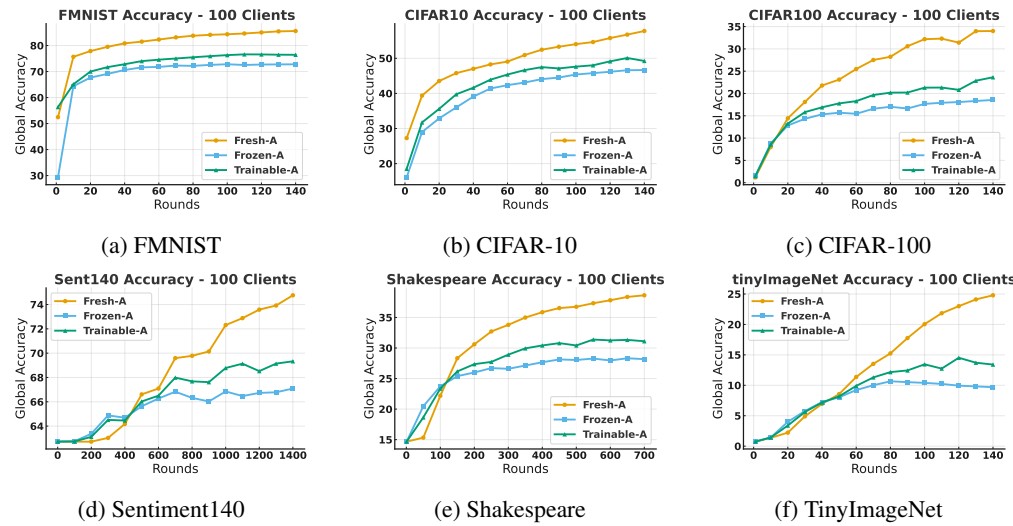

Figure 9: Global accuracy over communication rounds for six benchmarks under Fresh-A, Frozen-A, and Trainable-A. Trainable-A consistently stays between the two but remains closer to Frozen-A and fails to reach the performance of Fresh-A.

## G.1 ADDITIONAL EVALUATION ON FROZEN-A

To further test stability under increasing scale and complexity, we report two additional benchmarks, comparing FRESH-A with FROZEN-A and two intermediate schedules: **Frozen-First-50**, where $A$ is frozen for the first 50 rounds, then refreshed thereafter, and **Semi-Fresh-A**, where $A$ is refreshed every two rounds.

Table 13: **CelebA**: accuracy (%) over rounds for different $A$-schedules.

| Method | 0 | 10 | 20 | 30 | 40 | 50 | 60 | 70 | 80 | 90 | 100 |
|---|---|---|---|---|---|---|---|---|---|---|---|
| FRESH-A | 49.97 | 57.23 | 69.83 | 86.04 | 89.57 | 90.29 | 91.06 | 91.19 | 91.28 | 91.31 | 91.32 |
| SEMI-FRESH-A | 49.97 | 54.67 | 60.52 | 63.01 | 70.13 | 80.61 | 84.14 | 85.79 | 85.89 | 87.42 | 88.06 |
| FROZEN-FIRST-50 | 49.97 | 50.01 | 50.07 | 53.42 | 56.13 | 57.81 | 61.56 | 73.22 | 83.96 | 86.84 | 88.54 |
| FROZEN-A | 49.97 | 50.01 | 50.07 | 53.42 | 56.13 | 57.81 | 59.51 | 60.72 | 61.23 | 61.81 | 62.25 |

Table 14: **HAM10000**: accuracy (%) over rounds for different $A$-schedules.

| Method | 0 | 10 | 20 | 30 | 40 | 50 | 60 | 70 | 80 | 90 | 100 |
|---|---|---|---|---|---|---|---|---|---|---|---|
| FRESH-A | 67.99 | 68.13 | 69.08 | 70.41 | 70.68 | 71.32 | 72.26 | 72.75 | 73.17 | 73.79 | 74.13 |
| SEMI-FRESH-A | 67.99 | 68.08 | 68.44 | 69.77 | 69.86 | 69.54 | 70.44 | 70.75 | 71.03 | 71.12 | 70.96 |
| FROZEN-FIRST-50 | 67.99 | 67.99 | 67.99 | 67.99 | 67.99 | 67.99 | 68.21 | 69.38 | 70.07 | 71.04 | 72.97 |
| FROZEN-A | 67.99 | 67.99 | 67.99 | 67.99 | 67.99 | 67.99 | 67.99 | 67.99 | 67.99 | 67.99 | 67.99 |

**Takeaways.** (1) Across both datasets, FRESH-A remains stable in early rounds and attains the best final accuracy. (2) FROZEN-A exhibits limited improvement, consistent with a capacity-limited, fixed subspace. (3) Intermediate schedules (FROZEN-FIRST-50, SEMI-FRESH-A) offer smoother early phases than fully FRESH-A in a few cases but ultimately underperform the fully refreshed scheme. (4) As scale and complexity grow, exploration via a new round-wise subspace is beneficial for both stability and convergence quality.

## G.2 SENSITIVITY TO RANDOM SEEDS.

Across all of our experiments, MAPO shows very low sensitivity to the choice of random seed, as reflected in the standard deviations reported in Table 3. Since MAPO regenerates a new projection basis $A_t$ at every communication round, the effect of any individual "lucky" or "unlucky" random subspace is averaged out over the course of training. In our repeated trials with different seeds, the observed variance remains small and is comparable to the natural stochasticity of FedAvg. This agrees with our intuition that the exploration of many random subspaces prevents the training trajectory from being dominated by a single projection matrix.

To further isolate the role of the random seed, we conducted an expanded study on Frozen-A. This condition exposes the direct influence of the random seed on optimization dynamics. In this setting we observe a seed-dependent pattern:

- For **large subspaces** (for example, $k \geq d/2$), a fixed $A$ behaves similarly to full-rank training. Most seeds converge to comparable accuracies, although a small fraction of seeds underperform due to the omission of useful directions.
- For **small subspaces** (higher compression ratios), Frozen-A MAPO reliably converges to a suboptimal local minimum. The seed-to-seed variance becomes noticeably larger than in the Fresh-A case.
- In every such setting, **Fresh-A MAPO removes the majority of this variance** and delivers stable performance across seeds.

These findings align with the theoretical picture: large subspaces induce small projection error, so seeds matter mainly when the sampled subspace excludes important directions. Smaller subspaces induce larger projection error so a fixed subspace can trap optimization in seed-dependent minima. MAPO mitigates this issue by randomizing the projection basis across rounds, which averages out the variability and stabilizes convergence. Table 15 illustrates the observed pattern for 10 random seeds on the FMNIST dataset.

| Method | $k/d$ | Std | Min | Max |
|---------|---------|--------|--------|--------|
| Fresh-A | $1/2$ | 0.0021 | 0.8838 | 0.8900 |
| Frozen-A | $1/2$ | 0.0898 | 0.6720 | 0.8992 |
| Fresh-A | $1/100$ | 0.0035 | 0.8733 | 0.8836 |
| Frozen-A | $1/100$ | 0.0115 | 0.7389 | 0.7711 |

Table 15: Seed sensitivity of Fresh-A versus Frozen-A MAPO on FMNIST.

# H    NOTATIONS

Table 16: Notation and Definitions

| Symbol | Meaning / Definition |
|---|---|
| $N$ | Number of layers in a model. |
| $i$ | Indexing notation for the layers of the model. ($1 \le i \le N$) |
| $M$ | Number of clients in FL. |
| $j$ | Indexing notation for clients. ($1 \le j \le M$) |
| $T$ | Total number of communication rounds in FL. |
| $t$ | Indexing notation for rounds. ($1 \le t \le T$) |
| $\mathcal{D}^j$ | Local dataset for client $j$. |
| $b^j$ | Weight for client $j$, usually set as the number of local samples $|\mathcal{D}^j|$. |
| $\Delta W$ | Model update, treated as a single vector, $\in \mathbb{R}^{d \times 1}$. |
| $W^t$ | Model parameters at communication round $t$. |
| $\overline{B}^t$ | Aggregated projection vector at round $t$, broadcast by the server. |
| $r^t$ | Random seed used to synchronize matrix generation across clients and the server. |
| $A^t$ | Reconstruction matrix at round $t$, regenerated using $r_t$. |
| $B^{t,j}$ | Trainable projection matrix for client $j$ at round $t$. |
| $\hat{B}^{t,j}$ | Locally optimized projection matrix for client $j$ at round $t$. |
| $\eta$ | Learning rate for local optimization. |
| $d$ | Total number of model parameters, defined as $d = \sum_i d_1^i d_2^i$. |
| $d_1^i, d_2^i$ | Row and column dimensions of the weight matrix for layer $i$. |
| $p$ | Factorization rank after reshaping. |
| $q$ | LoRA Factorization rank before reshaping. |
| $k$ | Design parameter controlling reshape dimension ($\Delta W'$ reshaped into $\mathbb{R}^{\lceil d/k \rceil \times k}$). |
| $A \in \mathbb{R}^{\cdot \times \cdot}, B \in \mathbb{R}^{\cdot \times \cdot}$ | Reconstruction and projection matrices in factorization. |
| $\mathcal{L}(W)$ | Global loss function. |
| $\mathcal{L}^i(W)$ | Local loss function for client $i$. |
| $\nabla \mathcal{L}(W)$ | Gradient of the global loss function. |
| $\nabla B^{t,j}$ | Gradient of local loss for the projection matrix. |
| $\sigma_l^2$ | Bounded variance of stochastic gradients. |
| $\beta$ | Smoothness constant of the loss function. |
| $\epsilon$ | Distortion parameter from the Johnson-Lindenstrauss Lemma. |

# I PROOF OF DEFINITIONS AND PROPOSITIONS

**Definition I.1** (**Communication Overhead Rate**). *Let $\Delta W \in \mathbb{R}^{d_1 \times d_2}$ be the update matrix of a model. Suppose the factorization of $\Delta W$ as $\Delta W = BA$, where $A \in \mathbb{R}^{q \times d_2}$ is a fixed random matrix and $B \in \mathbb{R}^{d_1 \times q}$ is a trainable matrix with $q \le \min(d_1, d_2)$ being the factorization rank. The* **communication overhead rate** $\mathrm{CO}_{rate}$ *is defined as the ratio of the size of $B$ to the size of $\Delta W$:*

$$\mathrm{CO}_{rate} = \frac{\mathrm{size}(B)}{\mathrm{size}(\Delta W)} = \frac{q}{d_2}.$$

**Definition I.2** (**Reconstruction Error Rate**). *Using the same factorization as Definition 3.2, the reconstruction error rate is the expected ratio of the reconstruction error to the original model update. Given full-rank random reconstruction (Lemma 3.1), it is expressed as:*

$$\frac{\mathbb{E}_A \left[ \|\Delta W - BA\|_2^2 \right]}{\|\Delta W\|_2^2} = 1 - \frac{q}{d_2}.$$

*Proof.* Let $\Delta W = [\Delta w_1 \ \Delta w_2 \ \cdots \ \Delta w_{d_1}]$, where each column $\Delta w_i \in \mathbb{R}^{d_2}$. Similarly, the reconstruction $BA$ can be written as $[b_1 A \ b_2 A \ \cdots \ b_{d_1} A]$, where each $b_i \in \mathbb{R}^q$ is a trainable matrix. The reconstruction error is given by:

$$\|\Delta W - BA\|_2^2 = \sum_{i=1}^{d_1} \|\Delta w_i - b_i A\|_2^2.$$

The projection of $\Delta w_i$ onto the subspace spanned by $A$ is $P_A \Delta w_i$. The error rate $E$ is defined as:

$$E = \frac{\|\Delta w_i - \Delta w_i P_A\|_2^2}{\|\Delta w_i\|_2^2}.$$

Using the Pythagorean theorem:

$$\|\Delta w_i\|_2^2 = \|\Delta w_i P_A\|_2^2 + \|w_i - \Delta w_i \, P_A\|_2^2,$$

we rewrite $E$ as:

$$E = \frac{\|\Delta w_i\|_2^2 - \|\Delta w_i P_A\|_2^2}{\|\Delta w_i\|_2^2} = 1 - \frac{\|\Delta w_i P_A\|_2^2}{\|\Delta w_i\|_2^2}.$$

The expected value of $\|\Delta w_i P_A\|_2^2$ for a full-rank random Gaussian projection is:

$$\mathbb{E}[\|\Delta w_i P_A\|_2^2] = \frac{q}{d_2} \|\Delta w_i\|_2^2.$$

Substituting this into $E$:

$$\mathbb{E}[\|\Delta w_i - b_i A\|_2^2] = 1 - \frac{\mathbb{E}[\|\Delta w_i P_A\|_2^2]}{\|\Delta w_i\|_2^2} = 1 - \frac{\frac{p}{d} \|\Delta w_i\|_2^2}{\|w_i\|_2^2} = 1 - \frac{q}{d_2}.$$

Applying this to each column $\Delta \Delta w_i$ of $\Delta W$, we obtain:

$$\mathbb{E}_A \left[ \sum_{i=1}^{d_1} \|\Delta w_i - b_i A\|_2^2 \right] = \sum_{i=1}^{d_1} \mathbb{E}_A \left[ \|\Delta w_i - (\Delta w_i) P_A\|_2^2 \right].$$

Using the expected error formula:

$$= \sum_{i=1}^{d_1} \left( 1 - \frac{q}{d_2} \right) \|\Delta w_i\|_2^2 = \left( 1 - \frac{q}{d_2} \right) \sum_{i=1}^{d_1} \|\Delta w_i\|_2^2.$$

Since $\|\Delta W\|_2^2 = \sum_{i=1}^{d_1} \|\Delta w_i\|_2^2$, we get:

$$\mathbb{E}_A \left[ \|\Delta W - BA\|_2^2 \right] = \left( 1 - \frac{q}{d_2} \right) \|\Delta W\|_2^2.$$

$\square$

**Proposition I.3** (**Single-Vector Factorization**). *Let $\Delta W$, $A$, and $B$ be factorizations of a single layer of the network as in Definition 3.2. By reshaping $\Delta W$ into $\Delta W' \in \mathbb{R}^{1 \times d_1 d_2}$ the factorization of $\Delta W' = B'A'$ where $A' \in \mathbb{R}^{p \times d_1 d_2}$ and $B' \in \mathbb{R}^{1 \times p}$ can achieve the same **reconstruction error** and **communication overhead** to the conventional factorization of $\Delta W$ when $p = qd_1$.*

*Proof of Error Preservation.* In the single-vector setup, $\Delta W' \in \mathbb{R}^{d_1 d_2}$ is projected onto a subspace of dimension $p$. From random projection theory (as used in Definition 3.3), if $A'$ is sampled such that $\text{rank}(A') = p$, then:

$$\mathbb{E}\left[\frac{\|\Delta W' - B'A'\|_2^2}{\|\Delta W'\|_2^2}\right] = 1 - \frac{p}{d_1 d_2}.$$

Substituting $p = qd_1$ gives:

$$1 - \frac{qd_1}{d_1 d_2} = 1 - \frac{q}{d_2}.$$

Hence, the expected reconstruction error satisfies:

$$\mathbb{E}\left[\|\Delta W' - B'A'\|_2^2\right] = \left(1 - \frac{q}{d_2}\right)\|\Delta W'\|_2^2,$$

which matches the original factorization. $\qquad\square$

*Proof of Communication Preservation.* For $\Delta W' \in \mathbb{R}^{d_1 d_2}$, with the total size $\text{size}(\Delta W') = d_1 d_2$, we have the communication overhead as:

$$\text{size}(B') = p = qd_1.$$

Thus, the communication overhead is:

$$\text{CO}'_{rate} = \frac{\text{size}(B')}{\text{size}(\Delta W')} = \frac{qd_1}{d_1 d_2} = \frac{q}{d_2},$$

which matches the original overhead.

Since both the expected reconstruction error and the communication overhead remain unchanged, the single-vector factorization with $p = qd_1$ is equivalent in terms of efficiency. $\qquad\square$

**Proposition I.4** (**Multi-Layer Factorization**). *Let $\Delta W_i$, $A_i$, and $B_i$ be **single-vector factorization** of $i$-th layer of the $n$-layered network as in Proposition 3.4. By concatenating the reshaped weights $\Delta W_i$ into $\Delta W' \in \mathbb{R}^{1 \times d}$, where $d = \sum_{i=1}^{n} d_1^i d_2^i$. The factorization of $\Delta W' = B'A'$ where $A' \in \mathbb{R}^{p \times d}$ and $B' \in \mathbb{R}^{1 \times p}$ can achieve the same **reconstruction error** and **communication overhead** to the single-vector factorization applied to each $\Delta W_i$ when $p = nqd_1$.*

*Proof of Error Preservation.* For each layer $i$, a random full-rank matrix $A_i \in \mathbb{R}^{q \times d_2^i}$ yields an expected squared reconstruction error

$$\mathbb{E}\left[\|\Delta W_i - B_i A_i\|_F^2\right] = \left(1 - \frac{q}{d_2^i}\right)\|\Delta W_i\|_F^2.$$

Flattening $\Delta W_i$ into $\Delta W'_i \in \mathbb{R}^{(d_1^i d_2^i) \times 1}$, a single-vector projection of dimension $q\, d_1^i$ preserves this same error ratio (cf. Proposition 3.4).

When we concatenate all $\Delta W'_i$ into $\Delta W' \in \mathbb{R}^{1 \times d}$, we form a block-structured vector. Let $p := n\, q$ and let $A' \in \mathbb{R}^{p \times d}$ be constructed from a Gaussian distribution. By the standard random-projection argument in dimension $d$ with subspace size $p$,

$$\mathbb{E}\left[\|\Delta W' - B'A'\|_2^2\right] = \left(1 - \frac{p}{d}\right)\|\Delta W'\|_2^2 = \left(1 - \frac{p}{Nd_1 d_2}\right)\|\Delta W'\|_2^2.$$

Since $p = Nqd_1$, the overall ratio matches applying single-vector factorizations of rank $q$ to each $\Delta W'_i$ individually. $\qquad\square$

*Proof of Communication Preservation.* For each layer $i$, the single-vector factorization of $\Delta W_i$ introduces

$$\text{size}(B_i) = q\, d_1^i, \quad \text{size}(\Delta W_i) = d_1^i\, d_2^i, \quad \text{hence} \quad \frac{\text{size}(B_i)}{\text{size}(\Delta W_i)} = \frac{q}{d_1^i}.$$

Concatenating all $\Delta W_i'$ into $\Delta W' \in \mathbb{R}^{1 \times d}$ gives $\text{size}(\Delta W') = d$, with

$$d \;=\; \sum_{i=1}^{N} d_1^i\, d_2^i.$$

Meanwhile, in the multi-layer factorization, the new trainable vector $B' \in \mathbb{R}^{1 \times p}$ has

$$\text{size}(B') \;=\; p \;=\; N\, q.$$

Thus

$$\frac{\text{size}(B')}{\text{size}(\Delta W')} \;=\; \frac{N\, q}{\sum_{i=1}^{N} \left(d_1^i\, d_2^i\right)},$$

which matches the total overhead of $N$ individual rank-$q$ factorizations (one per layer) in aggregate. Consequently, the communication overhead rate is also preserved.

Since both the expected reconstruction error (per layer or in total) and the communication overhead remain the same, choosing $p = N\, q$ for $\Delta W'$ is equivalent to applying single-vector factorization of rank $q$ separately to each layer. $\qquad\square$

**Proposition I.5** (**MAPO Factorization**). *Let $\Delta W$, $A$, $B$, and rank $p$ be a multi-layer factorization of a network as defined in Proposition 3.5. By reshaping $\Delta W \in \mathbb{R}^{1 \times d}$ into $\Delta W' \in \mathbb{R}^{k \times \lceil d/k \rceil}$, and the factorization of $\Delta W' = B'A'$ where $A' \in \mathbb{R}^{1 \times \lceil d/k \rceil}$ and $B' \in \mathbb{R}^{k \times 1}$, we can achieve the same* **reconstruction error** *and* **communication overhead** *to the multi-layer factorization of $\Delta W$ when $k = p$, while reducing the size of reconstruction matrix by a factor of $k^2$.*

*Proof of Error Preservation.* Since $\Delta W \in \mathbb{R}^{1 \times d}$ is reshaped into $\Delta W' \in \mathbb{R}^{k \times \lceil d/k \rceil}$, we still have $\|\Delta W'\|_F^2 = \|\Delta W\|_2^2$. When $A' \in \mathbb{R}^{1 \times \lceil d/k \rceil}$ is a suitable random projection (and $B' \in \mathbb{R}^{k \times 1}$ is fit accordingly), the rank-1 subspace of dimension 1 within $\lceil d/k \rceil$ induces the known expected error ratio

$$\mathbb{E}\Big[\|\Delta W' - B'A'\|_F^2\Big] \;=\; \big(1 - \tfrac{1}{\lceil d/k \rceil}\big)\,\|\Delta W'\|_F^2,$$

since the ambient dimension is $k \times \lceil d/k \rceil \approx d$. By taking $k = p$, we obtain (via standard random-projection arguments) the matching error ratio $1 - p/d$, up to negligible rounding. Therefore:

$$\mathbb{E}\Big[\|\Delta W' - B'A'\|_F^2\Big] \;=\; \big(1 - \frac{p}{d}\big)\,\|\Delta W'\|_F^2,$$

$$\square$$

*Proof of Communication Preservation.* The matrix $B' \in \mathbb{R}^{k \times 1}$ has size $k$ in total. Meanwhile, $\Delta W' \in \mathbb{R}^{k \times \lceil d/k \rceil}$ has size $k \times \lceil d/k \rceil \approx d$. Thus

$$\frac{\text{size}(B')}{\text{size}(\Delta W')} \;=\; \frac{k}{\lceil d/k \rceil\, k} \;\approx\; \frac{k}{d} = \frac{p}{d}.$$

Setting $k = p$ matches the original ratio $\frac{p}{d}$ from $B \in \mathbb{R}^{p \times 1}$ in the multi-layer factorization. $\qquad\square$

*Proof of Memory Reduction by Factor $k^2$.* In standard rank-$p$ factorizations for $\Delta W \in \mathbb{R}^{1 \times d}$, one typically stores a $p \times d$ projection plus a $1 \times p$ vector, whose total size scales as $dp + p$. By contrast, $A' \in \mathbb{R}^{1 \times \lceil d/k \rceil}$ plus $B' \in \mathbb{R}^{k \times 1}$ has combined size $\lceil d/k \rceil + k$. When $k = p$, the ratio of these sizes can be shown to drop by a factor of approximately $k^2$. Hence, the approach allocates $k^2$ times less memory than a naive $p \times d$ plus $1 \times p$ arrangement. As $p = k$

$$\frac{dp + p}{\lceil d/k \rceil + k} \;=\; \frac{dk + k}{\lceil d/k \rceil + k} \;\approx\; \frac{d + 1}{d/k^2 + 1} \;\approx\; k^2$$

Thus, the factorization $\Delta W' = B'A'$ with $k = p$ exactly preserves the original rank-$p$ error and overhead while using $k^2$-fold less memory for reconstruction matrix $A$. $\qquad\square$

## J CONVERGENCE ANALYSIS PROOF

Let $\{\mathcal{L}^j\}_{j=1}^M$ be client objectives and $\mathcal{L}(W) := \frac{1}{M} \sum_{j=1}^M \mathcal{L}^j(W)$ the global objective. Denote by $W^t$ the global model at the beginning of communication round $t \in \{0, 1, \dots\}$ and by $E \in \mathbb{N}$ the number of local steps per round.

**Assumption J.1.** *For each $j$, $\mathcal{L}^j(v)$ is $\beta$-smooth, i.e., $\|\nabla\mathcal{L}^j(u) - \nabla\mathcal{L}^j(v)\| \leq \beta\|u-v\|$ for any $u$, $v$.*

**Assumption J.2.** *Variance of the stochastic gradient of $D^j$ is bounded for each client $j$, i.e.,*

$$\mathbb{E}\left[\left\|\nabla\mathcal{L}^j(W) - \widetilde{\nabla}\mathcal{L}^j(W)\right\|^2\right] \leq \sigma_l^2$$

**Assumption J.3.** *Bounded clients' gradient dissimilarity: $\frac{1}{M}\sum_{j=1}^M \left\|\nabla\mathcal{L}^j(W) - \nabla\mathcal{L}(W)\right\|^2 \leq \sigma_g^2$.*

**Assumption J.4.** *At each communication round $t$, the server selects a subset $S_t \subset [M]$ with $|S_t| = m < M$ clients uniformly and the sampling variance is $\sigma_{het}^2 = \rho\sigma_g^2$ where $\rho = \frac{M-m}{m(M-1)}$.*

During round $t$, each participating client $j \in S_t$ performs $E$ local steps indexed by $e \in \{0, 1, \dots, E-1\}$. We adopt the standard "virtual iterate" device: denote by $W^{t,0} = W^t$ the round-$t$ starting point and by $W^{t,e}$ the (virtual) state before local step $e$; all local gradients are evaluated at these virtual states and then aggregated centrally as if applied to $W^t$.

For each local step, client $j$ forms a MAPO-projected direction $B_e^{t,j} A_e^t$ approximating the stochastic gradient $\widetilde{\nabla}\mathcal{L}^j(W^{t,e})$. Define the per-step projection error

$$e_e^{t,j} := \widetilde{\nabla}\mathcal{L}^j(W^{t,e}) - B_e^{t,j} A_e^t.$$

Define the sampled averages

$$\widetilde{g}_e^t := \frac{1}{m} \sum_{j \in S_t} \widetilde{\nabla}\mathcal{L}^j(W^{t,e}), \qquad \bar{e}_e^t := \frac{1}{m} \sum_{j \in S_t} e_e^{t,j}.$$

Let $\eta_t > 0$ be the (server) stepsize. The aggregated update is

$$W^{t+1} = W^t - \eta_t \sum_{e=0}^{E-1} \widetilde{g}_e^t + \eta_t \sum_{e=0}^{E-1} \bar{e}_e^t. \tag{6}$$

**Johnson–Lindenstrauss (JL) property for MAPO.** Let $0 < \epsilon < 1$ be the JL distortion. With high probability (w.h.p.) the MAPO projection satisfies a norm preservation bound implying

$$\mathbb{E}\left[\|e_e^{t,j}\|^2\right] \leq \epsilon\,\mathbb{E}\left[\left\|\widetilde{\nabla}\mathcal{L}^j(W^{t,e})\right\|^2\right], \qquad \text{hence} \qquad \mathbb{E}\left[\|\bar{e}_e^t\|^2\right] \leq \epsilon\,\mathbb{E}\left[\|\widetilde{g}_e^t\|^2\right]. \tag{7}$$

**Variance decomposition under heterogeneity and sampling.** By Assumptions J.2 to J.4 and unbiased client sampling,

$$\mathbb{E}\left[\left\|\widetilde{g}_e^t - \nabla\mathcal{L}(W^{t,e})\right\|^2\right] \leq \sigma_l^2 + \sigma_g^2 + \sigma_{het}^2 \qquad \text{for all } t \text{ and } e. \tag{8}$$

**Theorem J.5.** *Let Assumptions J.1 to J.4 hold, and suppose $\eta_t \leq \frac{1-4\epsilon}{4\beta(1+\epsilon)}$. Then, after $T$ communication rounds each with $E$ local steps, the following bound holds:*

$$\frac{1}{4H_T} \sum_{t=0}^{T-1} \eta_t\,\mathbb{E}\left[\|\nabla\mathcal{L}(W^t)\|^2\right] \leq \frac{\mathbb{E}[\mathcal{L}(W^0)] - \mathcal{L}^*}{EH_T} + 2E(\epsilon + \beta + \beta\epsilon)\left(\sigma_l^2 + \sigma_g^2 + \sigma_{het}^2\right)\frac{1}{H_T}\sum_{t=0}^{T-1}\eta_t^2,$$

*where $H_T = \sum_{t=0}^{T-1}\eta_t$, $\epsilon$ is the JL distortion parameter, and $\mathcal{L}^*$ is the minimum of $\mathcal{L}(W)$.*

*Proof.* By $\beta$-smoothness of $\mathcal{L}$ and total expectation,

$$\mathbb{E}\left[\mathcal{L}(W^{t+1}) - \mathcal{L}(W^t)\right] \leq \mathbb{E}\left[\langle\nabla\mathcal{L}(W^t), W^{t+1} - W^t\rangle\right] + \frac{\beta}{2}\,\mathbb{E}\left[\|W^{t+1} - W^t\|^2\right]. \tag{9}$$

Substitute equation 6 and split the inner product into two terms:

$$\mathbf{E}_1 := \mathbb{E}\left[\left\langle\nabla\mathcal{L}(W^t), -\eta_t \sum_{e=0}^{E-1} \widetilde{g}_e^t\right\rangle\right], \qquad \mathbf{E}_2 := \mathbb{E}\left[\left\langle\nabla\mathcal{L}(W^t), \eta_t \sum_{e=0}^{E-1} \bar{e}_e^t\right\rangle\right].$$

**Bounding $\mathbf{E}_1$.** For each $e$, add and subtract $\nabla\mathcal{L}(W^{t,e})$, then use $\langle a, b\rangle = \frac{1}{2}(\|a\|^2 + \|b\|^2 - \|a-b\|^2)$ and smoothness to absorb $\|\nabla\mathcal{L}(W^t) - \nabla\mathcal{L}(W^{t,e})\|^2$ (the same step-size condition enforced later ensures nonpositivity of the resulting coefficient):

$$-\eta_t\,\mathbb{E}\big[\langle \nabla\mathcal{L}(W^t),\, \widetilde{g}_e^{\,t}\rangle\big] = -\eta_t\,\mathbb{E}\big[\langle \nabla\mathcal{L}(W^t),\, \nabla\mathcal{L}(W^{t,e})\rangle\big] - \eta_t\,\mathbb{E}\big[\langle \nabla\mathcal{L}(W^t),\, \widetilde{g}_e^{\,t} - \nabla\mathcal{L}(W^{t,e})\rangle\big]$$

$$\leq -\frac{\eta_t}{2}\,\mathbb{E}\big[\|\nabla\mathcal{L}(W^t)\|^2\big] - \frac{\eta_t}{2}\,\mathbb{E}\big[\|\nabla\mathcal{L}(W^{t,e})\|^2\big] + \eta_t\,\mathbb{E}\Big[\big\|\widetilde{g}_e^{\,t} - \nabla\mathcal{L}(W^{t,e})\big\|^2\Big].$$

Summing over $e = 0,\ldots, E-1$ and invoking equation 8 yields

$$\mathbf{E}_1 \leq -\frac{\eta_t}{2}\,E\,\mathbb{E}\big[\|\nabla\mathcal{L}(W^t)\|^2\big] - \frac{\eta_t}{2}\sum_{e=0}^{E-1}\mathbb{E}\big[\|\nabla\mathcal{L}(W^{t,e})\|^2\big] + \eta_t\,E\,\big(\sigma_l^2 + \sigma_g^2 + \sigma_{\text{het}}^2\big). \tag{10}$$

**Bounding $\mathbf{E}_2$.** Using $\langle a, b\rangle \leq \frac{1}{4}\|a\|^2 + \|b\|^2$, Jensen, equation 7, and equation 8,

$$\mathbf{E}_2 \leq \frac{\eta_t}{4}\,\mathbb{E}\big[\|\nabla\mathcal{L}(W^t)\|^2\big] + \eta_t\,\mathbb{E}\left[\Big\|\sum_{e=0}^{E-1}\bar{e}_e^{\,t}\Big\|^2\right]$$

$$\leq \frac{\eta_t}{4}\,\mathbb{E}\big[\|\nabla\mathcal{L}(W^t)\|^2\big] + \eta_t\,E\sum_{e=0}^{E-1}\mathbb{E}\big[\|\bar{e}_e^{\,t}\|^2\big]$$

$$\leq \frac{\eta_t}{4}\,\mathbb{E}\big[\|\nabla\mathcal{L}(W^t)\|^2\big] + \epsilon\,\eta_t\,E\sum_{e=0}^{E-1}\mathbb{E}\big[\|\widetilde{g}_e^{\,t}\|^2\big]$$

$$\leq \frac{\eta_t}{4}\,\mathbb{E}\big[\|\nabla\mathcal{L}(W^t)\|^2\big] + 2\epsilon\,\eta_t\sum_{e=0}^{E-1}\mathbb{E}\big[\|\nabla\mathcal{L}(W^{t,e})\|^2\big] + 2\epsilon\,\eta_t\,E\,\big(\sigma_l^2 + \sigma_g^2 + \sigma_{\text{het}}^2\big). \tag{11}$$

**Bounding the quadratic term.** From equation 6 and $\|a+b\|^2 \leq 2\|a\|^2 + 2\|b\|^2$,

$$\mathbb{E}\big[\|W^{t+1} - W^t\|^2\big] \leq 2\eta_t^2\,\mathbb{E}\left[\Big\|\sum_{e=0}^{E-1}\widetilde{g}_e^{\,t}\Big\|^2\right] + 2\eta_t^2\,\mathbb{E}\left[\Big\|\sum_{e=0}^{E-1}\bar{e}_e^{\,t}\Big\|^2\right]$$

$$\leq 2\eta_t^2\,E\sum_{e=0}^{E-1}\mathbb{E}\big[\|\widetilde{g}_e^{\,t}\|^2\big] + 2\eta_t^2\,E\sum_{e=0}^{E-1}\mathbb{E}\big[\|\bar{e}_e^{\,t}\|^2\big]$$

$$\leq 4\eta_t^2\sum_{e=0}^{E-1}\mathbb{E}\big[\|\nabla\mathcal{L}(W^{t,e})\|^2\big] + 4\eta_t^2\,E^2\big(\sigma_l^2 + \sigma_g^2 + \sigma_{\text{het}}^2\big)$$

$$+ 2\epsilon\,\eta_t^2\,E\sum_{e=0}^{E-1}\mathbb{E}\big[\|\widetilde{g}_e^{\,t}\|^2\big]$$

$$\leq 4(1+\epsilon)\eta_t^2\sum_{e=0}^{E-1}\mathbb{E}\big[\|\nabla\mathcal{L}(W^{t,e})\|^2\big] + 4(1+\epsilon)\eta_t^2\,E^2\big(\sigma_l^2 + \sigma_g^2 + \sigma_{\text{het}}^2\big). \tag{12}$$

Plugging equation 10, equation 11, and equation 12 into equation 9,

$$\mathbb{E}\big[\mathcal{L}(W^{t+1}) - \mathcal{L}(W^t)\big] \leq \left(-\frac{\eta_t}{2}E + \frac{\eta_t}{4}\right)\mathbb{E}\big[\|\nabla\mathcal{L}(W^t)\|^2\big]$$

$$+ \left(-\frac{\eta_t}{2} + 2\epsilon\,\eta_t + 2\beta(1+\epsilon)\eta_t^2\right)\sum_{e=0}^{E-1}\mathbb{E}\big[\|\nabla\mathcal{L}(W^{t,e})\|^2\big]$$

$$+ \eta_t E(\sigma_l^2 + \sigma_g^2 + \sigma_{\text{het}}^2) + 2\epsilon\,\eta_t E(\sigma_l^2 + \sigma_g^2 + \sigma_{\text{het}}^2)$$

$$+ 2\beta(1+\epsilon)\eta_t^2\,E^2(\sigma_l^2 + \sigma_g^2 + \sigma_{\text{het}}^2).$$

Choose

$$\eta_t \leq \frac{1-4\epsilon}{4\beta(1+\epsilon)}, \tag{13}$$

so the coefficient of $\sum_e \mathbb{E}\|\nabla\mathcal{L}(W^{t,e})\|^2$ is nonpositive. Since $E \geq 1$, the first coefficient is at most $-\frac{\eta_t}{4}$. Dropping the nonpositive term, we obtain

$$\mathbb{E}\big[\mathcal{L}(W^{t+1}) - \mathcal{L}(W^t)\big] \leq -\frac{\eta_t}{4}\mathbb{E}\big[\|\nabla\mathcal{L}(W^t)\|^2\big] + 2E(\epsilon + \beta + \beta\epsilon)\,\eta_t^2\,(\sigma_l^2 + \sigma_g^2 + \sigma_{\text{het}}^2). \quad (14)$$

### J.1 TELESCOPING, AVERAGING, AND FINAL BOUND

Summing equation 14 over $t = 0, \ldots, T-1$ and using $\mathcal{L}(W^T) \geq \mathcal{L}^*$,

$$\mathcal{L}^* - \mathbb{E}\big[\mathcal{L}(W^0)\big] \leq \sum_{t=0}^{T-1}\left(-\frac{\eta_t}{4}\mathbb{E}\big[\|\nabla\mathcal{L}(W^t)\|^2\big] + 2E(\epsilon + \beta + \beta\epsilon)\,\eta_t^2\,(\sigma_l^2 + \sigma_g^2 + \sigma_{\text{het}}^2)\right).$$

Rearranging and dividing by $EH_T$ with $H_T := \sum_{t=0}^{T-1}\eta_t$ yields

$$\frac{1}{4H_T}\sum_{t=0}^{T-1}\eta_t\,\mathbb{E}\big[\|\nabla\mathcal{L}(W^t)\|^2\big] \leq \frac{\mathbb{E}[\mathcal{L}(W^0)] - \mathcal{L}^*}{EH_T} + 2E(\epsilon + \beta + \beta\epsilon)(\sigma_l^2 + \sigma_g^2 + \sigma_{\text{het}}^2)\frac{1}{H_T}\sum_{t=0}^{T-1}\eta_t^2.$$

$$(15)$$

This is exactly the claimed bound.

$$\square$$

# K COMPLEXITY ANALYSIS AND MAPO FLEXIBILITY

Propositions 3.4 to 3.6 established that the factorization error of MAPO depends only on the size of the projection vector, regardless of how layers are reshaped or vectorized. In this section, we provide a detailed computational and memory analysis that complements the theoretical results in the main text. We show that MAPO not only matches the factorization accuracy of layer-wise methods under the same communication budget, but also achieves substantially lower computation and memory overhead in practice. We further compare MAPO with EvoFed and FedLoRU to highlight the efficiency advantages and flexibility of our design.

**Baseline Matrix Multiplication Costs.** We first recall standard matrix multiplication costs. Given matrices $A \in I\!R^{n \times m}$ and $B \in I\!R^{p \times n}$, the cost of computing $C = BA \in I\!R^{p \times m}$ and the memory needed to store all three matrices is

$$\text{Time}_{BA} = O(mnp), \quad \text{Memory}_{B+A} = O(nm + pn).$$

In what follows, we measure overhead relative to a standard forward and backward pass of the base model. Let $d$ denote the total number of model parameters,

$$d = \sum_{i=1}^{n} d_i^1 d_i^2,$$

and let $b$ denote the batch size. Assuming a standard forward and backward pass costs approximately $6bd$ FLOPs, which is linear in $d$, scaled by the batch size $b$.

## K.1 COMPUTATIONAL COMPLEXITY

**MAPO.** In MAPO, the full parameter vector is reshaped into a matrix $W \in I\!R^{k \times \lceil d/k \rceil}$. We factorize $W$ using a projection vector $A \in I\!R^{1 \times \lceil d/k \rceil}$ and a reconstruction vector $B \in I\!R^{k \times 1}$. Both projection and reconstruction reduce to a single matrix product with cost

$$O\big(\lceil d/k \rceil \times k \times 1\big) \approx O(d).$$

Each matrix-vector multiplication requires approximately one multiplication and one addition per element, hence the total cost for projection and reconstruction is $4d$ FLOPs per optimization step. Therefore, the relative computational overhead of MAPO is therefore

$$\frac{4d}{6bd} = \frac{2}{3b}.$$

This quantity does not depend on the model dimension $d$. For example, for $b = 32$ the overhead is about 2.1% and it decreases for larger batch sizes.

**EvoFed.** In EvoFed, The full model update is partitioned into $p$ parts and each factorized separately with population rank $q$. We can write it as $W \in \mathbb{R}^{q \times \lceil d/p \rceil}$ factorized by a projection vector $A \in \mathbb{R}^{q \times \lceil d/p \rceil}$ and a reconstruction vector $B \in \mathbb{R}^{p \times q}$. Both projection and reconstruction have cost:

$$O(\lceil d/p \rceil \times q \times p) = O(dq),$$

which is applied once per round rather than once per batch. If each client performs $e$ local epochs per round, the relative overhead becomes

$$\frac{4dq}{6bed} = \frac{2q}{3be}.$$

which is negligible for realistic values hyperparameter. For instance, if $b = 32, e = 200, p = \frac{d}{10}$, and $q = 10$ the overhead is around 0.1%. The overhead decreases for larger $b$ and $e$ while increases with number of population $q$ and is independent of number of partitions $p$.

**FedLoRU.** In FedLoRU, low-rank updates are performed in a layer-wise manner. Consider the $i$-th layer with parameter matrix $W_i \in \mathbb{R}^{d_i^1 \times d_i^2}$. FedLoRU introduces a low-rank factorization with rank $q$, using matrices

$$A_i \in \mathbb{R}^{q \times d_i^1}, \qquad B_i \in \mathbb{R}^{d_i^2 \times q}.$$

The corresponding layer-wise update involves matrix products of the form $B_i A_i$, which have cost

$$O\big(d_i^1 \times q \times d_i^2\big) = O\big(q d_i^1 d_i^2\big).$$

Summing over all layers, the total extra cost is

$$O\Big(q \sum_{i=1}^{n} d_i^1 d_i^2\Big) = O(qd).$$

The total number of additional FLOPs per batch due to FedLoRU is roughly $4qd$. Relative to the baseline cost $6bd$, the overhead is

$$\frac{4qd}{6bd} = \frac{2q}{3b}.$$

This quantity is independent of the model dimension $d$, but grows linearly with the FedLoRU rank $q$. For example, with $q = 4$ and $b = 32$, the overhead is about

$$\frac{2 \cdot 4}{3 \cdot 32} \approx 8.3\%.$$

Thus, for the same batch size and error tolerance, MAPO achieves a factor of $q$ reduction in computation overhead compared to FedLoRU.

### K.2 EMPIRICAL WALL-CLOCK MEASUREMENTS

All experiments were executed on a single NVIDIA RTX 3090 GPU with 24 GB memory. Table 17 reports the average client-side time per batch (in seconds) over the full training process for FedAvg, MAPO, EvoFed, and FedLoRU. The measured overheads are fully consistent with the theoretical analysis above.

Table 17: Average client-side wall-clock time per batch (in seconds) and relative overhead compared to FedAvg.

| Dataset | Batch | FedAvg | MAPO | Over | EvoFed | Over | FedLoRU | Over |
|---|---|---|---|---|---|---|---|---|
| MNIST | 32 | 5.277 | 5.347 | 1.33% | 5.280 | 0.06% | 5.439 | 3.07% |
| FMNIST | 32 | 5.277 | 5.356 | 1.50% | 5.283 | 0.11% | 5.468 | 3.62% |
| CIFAR10 | 32 | 47.393 | 48.173 | 1.64% | 47.435 | 0.09% | 50.143 | 5.80% |
| CIFAR100 | 32 | 67.705 | 68.750 | 1.54% | 67.724 | 0.03% | 71.139 | 5.07% |
| TinyImageNet | 32 | 117.965 | 119.409 | 1.22% | 118.044 | 0.06% | 130.517 | 10.64% |
| Shakespeare | 32 | 0.634 | 0.642 | 1.26% | 0.635 | 0.16% | 0.656 | 3.47% |
| Sent140 | 32 | 0.304 | 0.308 | 1.40% | 0.305 | 0.33% | 0.314 | 3.29% |
| SVHN | 32 | 32.478 | 32.961 | 1.49% | 32.490 | 0.04% | 34.061 | 4.87% |
| GLUE | 128 | 127.835 | 128.924 | 0.85% | 128.056 | 0.17% | 146.989 | 14.98% |

Across all datasets, the overhead of MAPO remains at or below two percent, EvoFed stays within a fraction of a percent, and FedLoRU incurs a significantly higher cost, especially on larger models such as TinyImageNet and GLUE. These measurements confirm that the additional computation required by MAPO is negligible in practice and does not offset its communication benefits.

### K.3 MEMORY COMPLEXITY OF MAPO, EVOFED, AND FEDLORU

**MAPO.** MAPO introduces two additional vectors, $A \in \mathbb{R}^{1 \times \lceil d/k \rceil}$ and $B \in \mathbb{R}^{k \times 1}$. Counting both parameters and their corresponding activations, the additional memory requirement is

$$2\left(\frac{d}{k} + k\right).$$

The base model parameters and their activations require approximately $2d$ units of memory. Therefore, the relative memory overhead of MAPO is

$$\frac{\frac{2d}{k} + 2k}{2d} = \frac{1}{k} + \frac{k}{d}.$$

For a typical choice $k = d/100$, this becomes

$$\frac{100}{d} + \frac{1}{100},$$

which is slightly above $1\%$ for a realistic model size.

**EvoFed.** In EvoFed the matrix $W \in \mathbb{R}^{p \times \lceil d/p \rceil}$ factorized by a projection vector $A \in \mathbb{R}^{q \times \lceil d/p \rceil}$ and a reconstruction vector $B \in \mathbb{R}^{p \times q}$. EvoFed does not store activations, therefore the total memory is

$$q(\frac{d}{p} + p)$$

and the relative memory overhead will be

$$\frac{q}{2}(\frac{1}{p} + \frac{p}{d}).$$

Given the practical values of $q = 10, p = \frac{d}{10}$ we get

$$5(\frac{10}{d} + \frac{1}{10}),$$

which is around $50\%$ for a realistic model size.

**FedLoRU.** For FedLoRU, each layer $i$ allocates additional low-rank matrices $A_i$ and $B_i$ with total parameter count

$$qd_i^1 + qd_i^2.$$

Under assumptions of $d_i^1 = d_i^2 = d_i$, and a fixed communication budget, MAPO and FedLoRU can be related through

$$k = q\sum_{i=1}^{n} d_i, \qquad d = \sum_{i=1}^{n} (d_i)^2.$$

Using these relations, we can calculate the memory for storing parameters and activations:

$$\text{Memory}_{\text{FedLoRU}} = 2(q\sum_{i=1}^{n} d_i + q\sum_{i=1}^{n} d_i) = 4k.$$

Therefore, the relative memory overhead of FedLoRU is

$$\frac{4k}{2d} = \frac{2k}{d}.$$

Which for the same choice of $k = d/100$, it becomes $2\%$ overhead.

## K.4 MAPO FLEXIBILITY

Suppose our neural network has $n$ layers. Let:

$$W_i \in \mathbb{R}^{d_i^1 \times d_i^2} \quad \text{for each layer } i = 1, \ldots, n.$$

Let $d = \sum_{i=1}^{n} d_i^1 d_i^2$ be the total number of parameters (i.e., the sum of the entries across all layers). Let

$$d_1 = \sum_{i=1}^{n} d_i^1.$$

In many treatments of LoRA, the main communication or factor-size bottleneck arises from a factor that scales linearly with $q \cdot d_i^1$.

**LoRA Factorization Per Layer.** LoRA factorizes each layer $W_i$ of dimension $d_i^1 \times d_i^2$ with a fixed rank $q$. Concretely,

$$W_i \approx W_i + B_i A_i, \qquad A_i \in \mathbb{R}^{q \times d_i^2}, \quad B_i \in \mathbb{R}^{d_i^1 \times q}.$$

The number of additional parameters introduced by each low-rank pair $(A_i, B_i)$ is

$$\underbrace{d_i^1 \cdot q}_{\text{size of } B_i} + \underbrace{q \cdot d_i^2}_{\text{size of } A_i} = q \left( d_i^1 + d_i^2 \right).$$

Summing over all $n$ layers,

$$\sum_{i=1}^{n} \left( d_i^1 \cdot q + q \cdot d_i^2 \right) = q \sum_{i=1}^{n} \left( d_i^1 + d_i^2 \right).$$

Therefore, we can write the communication cost as:

$$\text{Communication cost} \approx q \sum_{i=1}^{n} d_i^1 = q \, d_1.$$

Since $q$ must be an integer, we see that the communication overhead comes in integer multiples $d_1$, as:

$$\text{LoRA total communication} \in \{ q \, d_1 \mid q = 1, 2, \ldots \}.$$

**There is no way to select** a non-integer $q$. Hence communication budgets strictly between $d_1$ and $2 \, d_1$ (or between $q \, d_1$ and $(q + 1) d_1$) are not possible in layer-wise LoRA. Therefore, Any attempt to finely tune the communication or factor budget (e.g., to $1.5 \, d_1$) is disallowed by LoRA's integral-rank requirement. This **rigidity** is precisely what we seek to overcome in MAPO.

**MAPO Factorization.** MAPO flattens or reshapes all parameters into one large matrix and then performs a single low-rank factorization with rank 1. A simplified abstraction is:

1. Reshape $w_1, \ldots, w_n$ into a single matrix $W \in \mathbb{R}^{k \times \lceil d/k \rceil}$, where $d = \sum_{i=1}^{n} d_i^1 d_i^2$ is the total parameter count. 2. Factor $W \approx A \, B$, with

$$A \in \mathbb{R}^{1 \times \lceil d/k \rceil}, \quad B \in \mathbb{R}^{k \times 1},$$

Once all parameters are merged, MAPO can proportionally allocate any communication budget as $k$ can be selected freely.

$$\underbrace{\lceil d/k \rceil}_{\text{size of } A} + \underbrace{k}_{\text{size of } B}.$$

Therefore, we can write the total communication as:

$$\text{MAPO total communication} \in \{ k \mid k = 1, 2, \ldots \}.$$

This is particularly important in communication-efficient FL since viable solutions can be found with communication cost $k < d_1$ or $d_1 < k < 2d_1$, which architecture-dependent layer-wise factorization can not offer.

