# OpenReview forum: "Reshape-then-Factorize: Communication-Efficient FL via Model-Agnostic Projection Optimization"
_ICLR.cc/2026/Conference — Submitted to ICLR 2026_

### Official Review · Reviewer_DYrW · 2025-10-24

**Soundness:** 3
**Presentation:** 3
**Contribution:** 2
**Rating:** 6
**Confidence:** 4

**Summary:**

The paper proposes a model-agnostic factorization method MAPO. The method first flattens the whole model’s gradient so that it ignores the model architecture, and reshapes the flattened model gradient into a matrix, then factorize it as BA with a small trainable projection vector B and a shared reconstruction vector A.
The paper also provides theoretical analysis on factorization properties of MAPO, which states that MAPO achieves computation and communication efficiency. Empirically, MAPO is evaluated on vision, text, and GLUE fine-tuning, showing good empirical performance.

**Strengths:**

The idea is creative: it’s model-agnostic and applies a factorization to the flattened whole-model gradient, not layer by layer, which makes it broadly usable without architecture-specific tweaks. Thus, the method is easy to plug into different models.
The theory is straightforward and MAPO preserves reconstruction error and communication overhead compared to model-dependent factorizations.
The experiments make a practical point: accuracy stays close to FedAvg.

**Weaknesses:**

The discussion of the “low-rank averaging problem” is still an open question. Exact aggregation may not be important since the aggregation error can be larger than the error induced by low-rank averaging. It needs a further justification. Thus, the paper argues MAPO avoids certain pitfalls, but it doesn’t give a scientific underpinning like a theorem. A direct comparison between frozen-A and trainable-A would make the argument more convincing.
MAPO doesn’t naturally accommodate architecture-dependent adaptations, which are becoming a big deal in LoRA-style methods. For instance, LoRA variants often assign different ranks per layer based on task relevance; MAPO’s model-agnostic design makes that kind of per-layer customization harder.
The approach could also be valuable in centralized training, but that angle isn’t explored. (In line 136 the text says Figure 2 shows centralized MNIST training, yet the setup still uses federated algorithms.)

**Questions:**

Is there any reason MAPO uses whole flatten gradient for factorization? For MAPO, we can also divide whole flatten gradient into several parts and apply factorization for each part, and it would be a good direction to make flexibility.
What is the exact algorithm for MAPO fine-tuning? Is MAPO fine-tuning adding vec(BA) at each round? If so, does MAPO have a benefit over LoRA regarding flexibility (LoRA can be used as plug-in module)?

---

> ### Author Response · Authors · 2025-11-21
>
> We sincerely thank the Reviewer for the thoughtful and constructive assessment of our work, and we greatly appreciate your recognition of MAPO’s creative model-agnostic design, its broad plug-and-play applicability across architectures, the straightforward theoretical analysis, and the strong practical performance on vision and text tasks.
>
> ### Before addressing the individual questions, we would like to clarify the objective and methodology of MAPO which hopefully addresses most of your concerns and questions and removes the confusion with LoRA or PEFT:
>
> As stated in Table 1, Section 2.2 and Section 6 (Limitations), MAPO is **not** a PEFT or fine-tuning method. Our comparisons with LoRA-based techniques appear only in the Appendix to satisfy curiosity rather than as core baselines; our main experiments train all models **from scratch**, and the objective is strictly **communication efficiency**, not parameter-efficiency or fine-tuning.
>
> We emphasized that MAPO and LoRA target _different problems_.
>
> -   **MAPO** performs **gradient** factorization for **communication-efficiency** in **federated optimization**.
>
> -   **LoRA** performs **parameter** factorization for **parameter-efficiency** and **fine-tuning**.
>
>
> These two settings impose fundamentally different constraints:
>
> -   **LoRA must be layer-wise** because the forward pass depends on the decomposed parameters. Each layer’s nonlinearity and activation structure must be preserved, which forces per-layer factorization and often requires careful rank tuning.
>
> -   **MAPO does not modify the forward pass** at all; the full-rank model parameters remain intact. MAPO only factorizes the **backward gradient** updates, enabling a global low-rank representation that adapts across layers.
>
> We hope our clarifications help resolve any confusion and further demonstrate the contribution and applicability of MAPO.
>
> > **W1**: Low-rank averaging problem
>
> We appreciate this insightful point. As stated in Page 2, Line 70, the low-rank averaging issue arises because, in general
> $$
> \frac{1}{M}\sum_{j=1}^M B_i^j A_i^j \neq
> \left(\frac{1}{M}\sum_{j=1}^M B_i^j\right)
> \left(\frac{1}{M}\sum_{j=1}^M A_i^j\right),
> $$
> and this mismatch indeed occurs when the reconstruction matrices (A_i^j) differ across clients. However, as shown analytically in FA-LoRA [1] and EvoFed [2], when all clients share the _same_ reconstruction matrix,
> $$
> A_i^1 = A_i^2 = \cdots = A_i^M = A_i^*
> $$
>
> The low-rank averaging becomes _algebraically identical_ to averaging the full-rank update:
> $$
> \frac{1}{M}\sum_{j=1}^M B_i^j A_i^*
> = \left(\frac{1}{M}\sum_{j=1}^M B_i^j\right) A_i^*.
> $$
>
> MAPO inherits exactly this property. Even though MAPO refreshes $A$ _between rounds_, **within each round all clients use an identical** $A$for a shared random seed. Thus, where averaging occurs, MAPO satisfies the same condition under which the low-rank averaging error vanishes. In other words, the mismatch highlighted in Equation (1) does not arise in MAPO’s update rule because the aggregation is always performed over a common subspace basis.
>
> We updated Section 1 (“Challenges and Key Ideas”) to make this point more straightforward.

---

> ### Author Response · Authors · 2025-11-21
>
> > **W2**: a direct comparison between frozen-A and trainable-A
>
> A Trainable-A removes the possibility of communicationless synchronization of $A$, and we additionally introduce the low-rank averaging problem as you mentioned.
>
> Given the total communication rate for Frozen-A is $k$, Trainable-A demands the $d/k' + k'$ communication, which at best equals $2\sqrt{d}$ given $k'=\sqrt{d}$, while in practice, MAPO can apply rank $k < 2\sqrt{d}$ while matching a full-rank performance.
>
> For example, in GLUE tasks with 357million parameters, the $2\sqrt{d} = 37789$, while MAPO outperforms baseline results with $k=d/10,000=35,700$ and $k=d/100,000=3,570$.
>
> Similarly, as shown in Figure 6 for FMNIST and Shakespeare tasks, MAPO achieves similar results to FedAvg, by having $COM_{fmnist} = 32$ and $COM_{shakes}=256$ per round, while Trainable-A at least demands a $COM_{fmnist} = 213$ and $COM_{shakes}=1,806$.
>
> We anticipate that this additional communication might improve convergence rate and reduce the total number of rounds and communication cost; however, our experiments show that, while they converge faster in local training, the low-rank averaging problem makes global training unstable and prevents convergence.
>
> While we could fix this issue with full-rank aggregation at the server and full model transfer to clients, the final performance was still lower than Fresh-A MAPO, and much closer to Frozen-A MAPO, given the high communication rate, as shown in the following tables.
>
> We think that regenerating the $A$ matrix at each round can promote the exploration of new subspaces, especially with a minimal budget, better than gradient descent optimization of $A$.
> While this technique is effective on MAPO, LoRA-based architectures, or general parameter-factorization can not benefit from matrix $A$ regeneration, as the original model parameter is fixed.
>
> We believe this direction is worth exploring further in both gradient and parameter factorization; however, it is beyond the scope of our work, and we refer to it for future contributions.
>
> ---
> ### Sent140
>
> |Method|1|100|200|300|400|500|600|700|800|900|1000|1100|1200|1300|1400|
> |-|-|-|-|-|-|-|-|-|-|-|-|-|-|-|-|
> |Fresh-A|62.72|62.72|62.72|63.03|64.18|66.61|67.09|69.59|69.78|70.14|72.31|72.88|73.58|73.92|74.76|
> |Frozen-A|62.73|62.75|63.37|64.88|64.69|65.63|66.27|66.83|66.32|66.03|66.85|66.47|66.73|66.78|67.09|
> |Trainable-A|62.73|62.74|63.12|64.51|64.46|66.03|66.50|67.99|67.68|67.61|68.78|69.13|68.52|69.14|69.33|
>
> ---
> ### Shakespeare
>
> |Method|1|50|100|150|200|250|300|350|400|450|500|550|600|650|700|
> |-|-|-|-|-|-|-|-|-|-|-|-|-|-|-|-|
> |Fresh-A|14.67|15.33|22.18|28.33|30.60|32.71|33.79|34.99|35.86|36.54|36.75|37.33|37.82|38.36|38.63|
> |Frozen-A|14.69|20.47|23.73|25.40|25.99|26.71|26.60|27.13|27.67|28.13|28.04|28.27|27.96|28.32|28.18|
> |Trainable-A|14.68|18.64|23.35|26.17|27.36|27.73|28.91|29.94|30.42|30.80|30.42|31.39|31.25|31.33|31.11|
>
> ---
> ### tinyImageNet
>
> |Method|1|10|20|30|40|50|60|70|80|90|100|110|120|130|140|
> |-|-|-|-|-|-|-|-|-|-|-|-|-|-|-|-|
> |Fresh-A|0.74|1.37|2.22|4.90|6.95|8.55|11.37|13.52|15.24|17.76|20.05|21.84|23.00|24.11|24.79|
> |Frozen-A|0.75|1.43|4.00|5.77|7.25|8.03|9.22|10.01|10.66|10.52|10.40|10.23|9.94|9.82|9.69|
> |Trainable-A|0.75|1.42|3.39|5.62|7.17|8.18|9.91|11.30|12.17|12.44|13.45|12.75|14.53|13.71|13.41|
>
> ---
> ### CIFAR-10
> |Method|1|10|20|30|40|50|60|70|80|90|100|110|120|130|140|
> |-|-|-|-|-|-|-|-|-|-|-|-|-|-|-|-|
> |Fresh-A|27.26|39.38|43.52|45.77|47.01|47.95|48.69|49.21|49.73|50.23|50.67|50.94|51.09|51.15|51.20|
> |Frozen-A|16.08|29.00|32.87|36.01|39.08|41.34|43.08|44.51|45.72|46.73|47.53|48.14|48.57|48.83|49.01|
> |Trainable-A|18.51|31.73|35.60|39.74|41.62|43.38|44.90|46.08|47.05|47.91|48.60|49.09|49.47|49.78|49.97|
>
> ---
> ### CIFAR-100
> |Method|1|10|20|30|40|50|60|70|80|90|100|110|120|130|140|
> |-|-|-|-|-|-|-|-|-|-|-|-|-|-|-|-|
> |Fresh-A|1.19|8.01|14.44|18.11|21.79|23.12|25.49|27.51|28.23|30.60|32.18|32.30|31.41|33.97|34.01|
> |Frozen-A|1.55|8.79|12.83|14.33|15.32|15.69|15.45|16.58|17.02|16.63|17.67|17.92|18.08|18.32|18.55|
> |Trainable-A|1.46|8.57|13.25|15.82|16.90|17.78|18.28|19.66|20.19|20.21|21.31|21.33|20.82|22.87|23.64|
>
> ---
> ### FMNIST
> |Method|1|10|20|30|40|50|60|70|80|90|100|110|120|130|140|
> |-|-|-|-|-|-|-|-|-|-|-|-|-|-|-|-|
> |Fresh-A|52.46|75.67|77.88|79.51|80.84|81.51|82.31|83.15|83.77|84.12|84.34|84.64|85.04|85.45|85.59|
> |Frozen-A|29.30|64.38|67.68|69.10|70.60|71.61|71.78|72.30|72.19|72.56|72.81|72.53|72.70|72.75|72.78|
> |Trainable-A|56.34|65.12|70.01|71.70|72.83|74.01|74.57|75.03|75.49|75.93|76.32|76.61|76.58|76.48|76.44|
>
> ---
> We added these new findings and visual plots of the tables to the revised Appendix G, where previously only Fresh-A vs. Frozen-A had been studied.

---

> ### Author Response · Authors · 2025-11-24
>
> > **W3**: MAPO doesn’t naturally accommodate architecture-dependent adaptations, which are becoming a big deal in LoRA-style methods. For instance, LoRA variants often assign different ranks per layer based on task relevance; MAPO’s model-agnostic design makes that kind of per-layer customization harder.
>
> We appreciate this important observation. We would like to clarify that the absence of per-layer rank allocation is not a limitation of MAPO, but rather an intentional design choice that provides a key _advantage_ over layer-wise factorization methods.
>
> Layer-wise low-rank approaches (including many LoRA variants) assign a fixed rank, and thus a fixed update/communication budget, to each layer. This allocation is static and does not adapt to the amount of useful gradient signal in each layer at a given batch. In practice, layers contribute very unequally across batches, so forcing an identical or manually-tuned per-layer rank is often suboptimal.
>
> In contrast, MAPO factorizes the **full-model gradient**. The projection vector (B_t) automatically concentrates its limited capacity across all layers in whichever directions contribute the most to reducing the loss at that step. This yields a dynamic and data-dependent allocation of effective rank across layers, without manual tuning. The empirical difference between MAPO and FedLoRU already hints at this benefit that MAPO can shift capacity toward high-signal layers on the fly, whereas layer-wise methods cannot reallocate rank once chosen.
>
> Therefore, what seems like a lack of layer-specific customization in MAPO is actually what enables MAPO to (i) avoid heavy hyperparameter tuning, (ii) avoid architecture-dependent design, and (iii) dynamically reassign its limited projection budget to the most informative gradient directions at each step.
>
> In summary, LoRA’s per-layer static rank is a structural limitation and a requirement of **parameter-factorization** or **layer-wise** methods, whereas MAPO’s **global factorization** is a strength that enables more flexible, automatically adaptive use of the low-rank budget.
>
> > **W4**: MAPO in centralized training
>
> Thank you for pointing this out. The intention of Figure 2 was to provide a controlled setting to isolate factorization behavior from data distribution and aggregation, not to propose MAPO as a centralized-training method.
>
> In essence, MAPO in a centralized setting does not bring additional benefits: it incurs higher computational and memory costs (around 2% for a given rank $k = \frac{d}{100}$). It yields a low-rank approximation of the high-quality gradients.
>
> While low-rank approximation can still converge and train the model, it slows convergence and is most beneficial for improving communication efficiency in distributed settings rather than centralized training.
>
> As we stated before, the MAPO objective is not PEFT or parameter updates; therefore, it is better suited for addressing problems in communication efficiency in Federated learning or Federated full-rank fine-tuning.
>
> > **Q1**:Is there any reason MAPO uses a whole flatten gradient for factorization? For MAPO, we can also divide the whole flatten gradient into several parts and apply factorization for each part, and it would be a good direction to make flexibility.
>
> Great question. This is mathematically equivalent to what the reshaping step in MAPO does, if all the divided parts use the same reconstruction matrix $A$.
>
> Allowing each part has its own reconstruction matrix $A$, results in $k$ times more memory consumption, and since all $A$ matrices are randomly generated, they will not bring any useful correlation with the model gradients, therefore it does not matter if we use the same reconstruction matrix $A$ for all parts or we pay for additional $k$ times more memory and have a unique reconstruction for each part.
>
> This follows from Proposition 3.6, which shows that it has the same expected reconstruction error as Proposition 3.5, even when the same reconstruction matrix $A$ is repeatedly used instead of a unique one for each part.
>
> Additionally, the reshaping enables more effective parallelization of factorization within a single GPU, rather than performing parts factorization one by one.
>
> > **Q2**: Is MAPO fine-tuning adding vec(BA) at each round?
>
> We appreciate the question. Yes, at the end of each round, the full model parameter gets updated by $vec(BA)$.
>
> As stated in the main text and Table 1, MAPO is not a PEFT method. However, it can be used as a full-rank fine-tuning mechanism explicitly designed for communication efficiency, applicable to any model architecture (even alongside LoRA) without modifying the forward pass.

---

> ### Author Response · Authors · 2025-11-24
>
> > **Q3**: MAPO fine-tuning and comparison to LoRA
>
> As stated in the main text and Table 1, MAPO is not a PEFT method; therefore, it should not be compared to LoRA on the basis of parameter efficiency. MAPO is a full-rank fine-tuning mechanism explicitly designed for communication efficiency, and it can be applied to any model architecture without modifying the forward pass.
>
> In Appendix B (Table 4), we report the number of trainable parameters for MAPO full-rank fine-tuning. MAPO uses the full set of model parameters, while LoRA uses a much smaller number. This reflects the different objectives: MAPO optimizes full model parameters with low-rank gradients, while LoRA optimizes a small parameter subspace.
>
> In Appendix B (Table 5), we focus on communication efficiency. MAPO can outperform LoRA at a much smaller communication budget, since clients only transmit the low-rank projected gradient. Although MAPO is not parameter efficient, it is communication efficient and can achieve competitive or better accuracy than LoRA when communication is the primary bottleneck.
>
> The two approaches are compatible. MAPO is model agnostic and operates only on gradients, so it can be used with LoRA. A combined MAPO plus LoRA setup can provide both parameter efficiency (from LoRA) and communication efficiency (from MAPO). This hybrid direction is interesting but outside the scope of our work.
>
> The comparisons in Appendix B and Appendix C were included only to illustrate how gradient factorization (MAPO) differs from parameter factorization (LoRA, Factorized-FL) in terms of communication cost, not to present MAPO as an alternative PEFT method.

---

> > ### Author Response · Authors · 2025-11-28
> >
> > Thank you very much for taking the time to review our work and provide your feedback. We hope our answers are satisfactory. If you have any further questions or concerns, we would be more than happy to respond at any time.

---

### Official Review · Reviewer_p3zB · 2025-10-30

**Soundness:** 3
**Presentation:** 3
**Contribution:** 3
**Rating:** 4
**Confidence:** 3

**Summary:**

This paper proposes a novel federated learning method called Model Independent Projection Optimization (MAPO) to address the bottleneck problem of excessive communication overhead in federated learning. Unlike traditional layer-by-layer low-rank decomposition methods, MAPO reshapes and decomposes the gradient of the entire model into a fixed reconstruction matrix and a trainable projection vector, thereby freeing it from model architecture constraints and achieving a more flexible low-rank representation.

**Strengths:**

1. The proposed MAPO method fundamentally changes the paradigm of traditional low-rank decomposition, avoiding architectural constraints through model-level decomposition and providing a novel approach to communication efficiency optimization.

2. The paper provides rigorous theoretical analysis, including definitions of communication overhead rate and reconstruction error rate, as well as mathematical proofs and convergence analyses of MAPO decomposition, enhancing the credibility of the method.

**Weaknesses:**

1. Insufficient computational complexity analysis: While the paper mentions that MAPO is more efficient than layer-by-layer methods, the specific analysis of actual computational overhead is not in-depth enough, and there is a lack of practical performance evaluation on resource-constrained devices.

2. Inadequate discussion of hyperparameter sensitivity: The choice of parameter k has a significant impact on performance, but the paper does not provide guidelines for choosing the system's k value, which may increase the difficulty of tuning in practical deployments.

3. Insufficient security considerations: The paper does not discuss the potential security risks associated with the generation and synchronization of the dynamic reconstruction matrix in privacy-sensitive federated learning scenarios.

**Questions:**

See Weaknesses

---

> ### Author Response · Authors · 2025-11-21
>
> We sincerely thank the reviewer for the thoughtful and constructive evaluation of our work. We truly appreciate the recognition of MAPO as a novel shift in the paradigm of low rank decomposition, its ability to remove architectural constraints through full model-level reshaping, and its contribution to communication efficiency in federated learning. We are also grateful for the positive assessment of our theoretical analysis and the convergence proofs.
>
> ---
>
> > **W2:** Inadequate discussion of hyperparameter sensitivity
>
> Thank you for highlighting the importance of $k$ sensitivity. In Section 6 (MAPO Hyperparameter), we directly addressed your concern about $k$ sensitivity and the guidelines for hyperparameter tuning in this study.
> Figure 6 illustrates the sensitivity for the FMNIST and Shakespeare datasets, and Appendix B and C showcase the sensitivity to $k$ on GLUE, SVHN, and CIFAR-10, and report the trade-off between performance and communication load alongside other baselines.
>
> We revised Section 6 (MAPO Hyperparameter) and Appendix D.1 to include the following more detailed tuning approach, which we used to search for the best $k$.
>
> To choose an optimal value of $k$, we begin by evaluating MAPO under a high compression setting, typically starting around $k \approx \sqrt{d}$. We then increase $k$ progressively by doubling its value at each step. After each evaluation, we compare both the final accuracy and the convergence rate. The search continues until we observe that increases in $k$ no longer yield meaningful performance gains, and the faster convergence no longer compensates for the higher communication cost. In practice, this means choosing the smallest $k$ that achieves near-optimal accuracy while also minimizing the total number of communication rounds.
>
> For example, if $k=128$ reaches 90 percent accuracy in 100 rounds and $k=256$ reaches the same accuracy in 60 rounds, the choice of $k=128$ remains preferable because the overall communication load is lower despite the slower convergence.
>
> Many other random or Bayesian searches can be used; while the computation is fixed for any $k$, the memory increase is linear, therefore it is more feasible to search the lower bounds faster in parallel and stop as the first satisfactory $k$.
>
> > **W3:** Insufficient security considerations: The paper does not discuss the potential security risks associated with the generation and synchronization of the dynamic reconstruction matrix in privacy-sensitive federated learning scenarios.
>
> Thank you for the question. The generation and synchronization of the dynamic reconstruction matrix happen without any need for communication with a public seed value.
> Therefore, it is always local and no attackers can tamper with the generation or synchronization of the reconstruction matrix $A$.
>
> The matrix $A$ also does not carry any meaningful information about the client data or the state of training; therefore, publicly and insecurely transferring seeds does not reveal any information that attackers can exploit, as long as the projection vector $B$ is secure.
>
> Malicious tampering with the projection vector $B$  is algebraically equivalent to gradient poisoning; therefore, Byzantine-robust aggregation methods [1, 2, 3] are directly applicable.
>
> Server-side errors resemble standard FL model update faults, for which established defenses, such as secure aggregation [4], or using a Fully Homomorphic Encryption (FHE) and third-party Trusted Execution Environments (TEEs) as in EvoFed [5], remain fully compatible. Notably, MAPO’s low-dimensional update vector substantially reduces both communication and computation costs associated with secure aggregation.
>
> In summary, MAPO seamlessly integrates with existing robustness strategies in FL without requiring modification or introducing new risks.
>
>
>
>
> ```
> [1] Yin, D. et. al (2018, July). Byzantine-robust distributed learning: Towards optimal statistical rates. In _International conference on machine learning_ (pp. 5650-5659). Pmlr.
> [2] Blanchard, P., et. al (2017). Machine learning with adversaries: Byzantine tolerant gradient descent. _Advances in neural information processing systems_, _30_.
> [3] Zhu, B. et. al (2023, April). Byzantine-robust federated learning with optimal statistical rates. In _International Conference on Artificial Intelligence and Statistics_ (pp. 3151-3178). PMLR.
> [4] Bonawitz, K.et. al (2017, October). Practical secure aggregation for privacy-preserving machine learning. In _proceedings of the 2017 ACM SIGSAC Conference on Computer and Communications Security_ (pp. 1175-1191).
> [5] Rahimi, M. M. et. al (2023). EvoFed: leveraging evolutionary strategies for communication-efficient federated learning. Advances in Neural Information Processing Systems, 36, 62428-62441.
> ```

---

> ### Author Response · Authors · 2025-11-21
>
> > **W1:** Insufficient computational complexity analysis
>
> We thank the reviewer for raising this concern. As clarified in our revised Appendix K, the additional computation introduced by MAPO is indeed present, but it remains marginal in practice and is significantly lower than that of layer-based low-rank methods. Below, we summarize the key computational and memory results now included in the revised Appendix.
>
> MAPO performs reconstruction and projection using a single matrix multiplication. The full gradient vector is reshaped into $W \in \mathbb{R}^{k \times \lceil d/k \rceil}$ and factorized by a projection vector $A \in \mathbb{R}^{1 \times \lceil d/k \rceil}$ and a reconstruction vector $B \in \mathbb{R}^{k \times 1}$. Both projection and reconstruction have cost $O(\lceil d/k \rceil \times k \times 1) = O(d)$. Since each operation requires one multiplication and one addition per element, the total FLOPs per optimization step is $4d$. Relative to the cost of a standard forward plus backward pass, approximately $6bd$, MAPO's overhead becomes $\frac{4d}{6bd} = \frac{2}{3b}$, which is independent of $d$ and remains near $2.1$% for $b=32$ and decreases further for larger $b$.
>
> In EvoFed, the full model update is partitioned into $p$ parts and each factorized separately with population rank $q$. We can write it as $W \in \mathbb{R}^{p \times \lceil d/p \rceil}$ factorized by a projection vector $A \in \mathbb{R}^{q \times \lceil d/p \rceil}$ and a reconstruction vector $B \in \mathbb{R}^{p \times q}$.
> Both projection and reconstruction have cost $O(\lceil d/p \rceil \times q \times p) = O(dq),$
> which is applied once per round rather than once per batch. If each client performs \(e\) local epochs per round, the relative overhead becomes $\frac{4dq}{6bed} = \frac{2q}{3be}$. Which is negligible for realistic values of the hyperparameter. For instance, if $b=32, e=200, p=\frac{d}{10}$, and $q=10$ the overhead is around $0.1$%. The overhead decreases for larger $b$ and $e$, increases with the population size $q$, and is independent of the number of partitions $p$.
>
> For FedLoRU, the additional computation is performed layer-wise. If the $i$th layer has shape $W_i \in \mathbb{R}^{d_i^1 \times d_i^2}$ and rank $q$, the layer update requires $O(d_i^1 \times q \times d_i^2)$ FLOPs. Summed across layers, this becomes $O(qd)$, yielding an additional $4qd$ FLOPs per batch. Relative to $6bd$, the overhead is $\frac{4qd}{6bd} = \frac{2q}{3b}$, which grows linearly with $q$. For example, $q=4$ and $b=32$ produce about $8.3$% overhead. This shows that MAPO provides a $q$-factor reduction in computational cost compared to FedLoRU.
>
> Empirical wall-clock measurements show that MAPO's overhead remains below two percent across all datasets. EvoFed remains within fractions of a percent, while FedLoRU incurs significantly higher, particularly for larger models such as TinyImageNet and GLUE. These measurements validate that MAPO's computation overhead is negligible in practical training settings.
>
> |Dataset|Batch size|FedAvg|MAPO|Over Head|EvoFed|Over Head|FedLoRU|Over Head|
> |-|-|-|-|-|-|-|-|-|
> |MNIST|32|5.277|5.347|1.33%|5.28|0.06%|5.439|3.07%|
> |FMNIST|32|5.277|5.356|1.50%|5.283|0.11%|5.468|3.62%|
> |C10|32|47.393|48.173|1.64%|47.435|0.09%|50.143|5.80%|
> |C100|32|67.705|68.75|1.54%|67.724|0.03%|71.139|5.07%|
> |Tiny|32|117.965|119.409|1.22%|118.044|0.06%|130.517|10.64%|
> |Shak|32|0.634|0.642|1.26%|0.635|0.16%|0.656|3.47%|
> |Sent|32|0.304|0.308|1.40%|0.305|0.33%|0.314|3.29%|
> |SVHN|32|32.478|32.961|1.49%|32.49|0.04%|34.061|4.87%|
> |GLUE|128|127.835|128.924|0.85%|128.056|0.17%|146.989|14.98%|
>
> MAPO introduces two vectors, $A \in \mathbb{R}^{1 \times \lceil d/k \rceil}$ and $B \in \mathbb{R}^{k \times 1}$, whose parameter plus activation memory is $2(\frac{d}{k} + k)$. Since the base model requires approximately $2d$ units of memory, the relative memory overhead of MAPO is $\frac{1}{k} + \frac{k}{d}$. For a typical choice such as $k=d/100$, this evaluates to slightly above $1$%.
>
> In **EvoFed** the matrix $W \in \mathbb{R}^{p \times \lceil d/p \rceil}$ factorized by a projection vector $A \in \mathbb{R}^{q \times \lceil d/p \rceil}$ and a reconstruction vector $B \in \mathbb{R}^{p \times q}$. EvoFed does not store activations, therefore the total memory is $q(\frac{d}{p}+p)$ and the relative memory overhead will be $\frac{q}{2}(\frac{1}{p}+\frac{p}{d})$. Given the practical values of $q=10, p=\frac{d}{10}$ we get $5(\frac{10}{d} + \frac{1}{10}) \approx 50$%.
>
> In contrast, **FedLoRU** introduces per layer low rank matrices with total parameter count $q d_i^1 + q d_i^2$. Under the simplifying assumption $d_i^1=d_i^2=d_i$ and a matched communication budget with MAPO, where $k = q\sum_i d_i$, the total memory for parameters and activations becomes $4k$.
>
> The relative overhead is then $\frac{4k}{2d} = \frac{2k}{d}$, which corresponds to $2$% when $k = d/100$.
>
> These new results are included in Appendix K of the revised manuscript.

---

> > ### Author Response · Authors · 2025-11-28
> >
> > Thank you very much for taking the time to review our work and provide your feedback. We hope our answers are satisfactory. If you have any further questions or concerns, we would be more than happy to respond at any time.

---

### Official Review · Reviewer_91yB · 2025-10-31

**Soundness:** 2
**Presentation:** 2
**Contribution:** 1
**Rating:** 2
**Confidence:** 5

**Summary:**

The paper proposes MAPO (Model-Agnostic Projection Optimization), a method for improving communication efficiency in federated learning (FL). It aims to replace layer-wise low-rank or LoRA-style decompositions with a model-wide projection. Instead of constraining each layer’s gradients separately, MAPO reshapes the entire model gradient into a single matrix and factorizes it. The paper provides some FL based convergence analysis, and empirical evaluations on multiple datasets. The results show that MAPO achieves comparable accuracy to FedAvg while reducing communication cost.

**Strengths:**

1. The problem motivation is valid: communication efficiency is a critical bottleneck in large-scale FL.
2. The idea of global gradient projection using shared random seeds is simple and elegant; it generalizes many existing gradient compression approaches under a unified formulation.
3. The paper provides a formal convergence bound under standard smoothness and bounded-variance assumptions.
4. The experiments are broad, spanning image and text datasets with different architectures.

**Weaknesses:**

1. The novelty is extremely limited. MAPO essentially repackages global random projection with seed synchronization, ideas existing in one form or another in earlier methods. The contribution lies mainly in notation and aggregation formulation rather than a new algorithmic principle.
2. The comparison set is largely incomplete: several recent or highly relevant works are missing, such as FedEx-LoRA [1], SparseFL [2], FedRPCA [3], PRISM [4], Heterogeneous LoRA for Federated Fine-tuning [5], etc. These should have been discussed and compared empirically.
3. There is no rigorous analysis of computational cost. I understand that communication is an important bottleneck, but one cannot ignore the limited computational resources at the clients, and potentially at the server. It will be ideal to provide a rigorous analysis on the pareto front for computation-communication burden of the proposed scheme, with clear comparisons to prior works. The claim of efficiency is only on communication, while the proposed reshaping and projection steps add nontrivial local computation and memory overheads (e.g., matrix inversion or vector reshaping costs are ignored).
4. The convergence theory is largely boilerplate, relying on standard FL proofs without any MAPO-specific insight (the same result could hold for many gradient projection schemes). If there are three layers in a neural network, how does MAPO converge differently from doing a projection for each layer separately?
5. The paper’s writing is verbose, and the presentation sometimes obscures what is genuinely new. For instance, the reshaping step is more of a mathematical convenience than a conceptual innovation.

[1] FedEx-LoRA: Exact Aggregation for Federated and Efficient Fine-Tuning of Foundation Models. (ACL 2025)
[2] Revisiting Sparsity Hunting in Federated Learning: Why does Sparsity Consensus Matter? (TMLR 2023)
[3] FedRPCA: Enhancing Federated LoRA Aggregation Using Robust PCA. (arXiv)
[4] Overcoming Resource Constraints in Federated Learning: Large Models Can Be Trained with only Weak Clients. (TMLR 2023)
[5] Heterogeneous LoRA for Federated Fine-tuning of On-Device Foundation Models. (ACL 2024)

**Questions:**

Please address the weaknesses.

---

> ### Author Response · Authors · 2025-11-21
>
> We sincerely thank the reviewer for the assessment of our work, and we appreciate your recognition of several strengths, including the valid motivation, the simplicity and elegance of the global projection mechanism with shared random seeds and its ability to unify many existing gradient-compression approaches, the inclusion of a formal convergence bound, and the breadth of our experiments across tasks and architectures.
>
> ---
>
> ### Before addressing the individual questions, we would like to clarify the objective and methodology of MAPO that hopefully address most of your concerns and questions and removes the confusion with LoRA or PEFT:
>
>
> As stated in Table 1, MAPO is **not** a PEFT or fine-tuning method, and our comparisons with LoRA-based techniques appear only in the Appendix to satisfy curiosity rather than as core baselines; our main experiments (except GLUE) train all models **from scratch**, and the objective is strictly **communication efficiency**, not parameter-efficiency or personalized fine-tuning.
>
> > **W1**: The novelty is extremely limited.
>
> MAPO is also **not** a repackaging of existing low-rank projection methods: prior approaches either
> 1) rely on _layer-wise_ factorizations, which inherently prevent model-agnostic projection;
> 2) employ _global_ projections whose reconstruction matrices often requiring **k×** the model size, impractical for real deployments;
> 3) Compute the projection coefficients **post-hoc** via matrix multiplication, without learning them through gradient descent during local training. In contrast, MAPO treats the **projection vector of the full-model gradient** as a trainable variable, directly updated by SGD, enabling the low-rank subspace to adapt to loss-sensitive directions rather than serving as a post-training compression mechanism.
>
> > **W5**:  The reshaping step is more of a mathematical convenience...
>
> The reshaping step is therefore not merely a notational convenience, but the key mechanism that enables the entire model to share a common adaptive subspace with the _same reconstruction power_ as layer-wise factorizations under equal communication budgets (as proven in the Proposition 3.3-3.6).
>
> The results in Appendix K show that this reshaping not only preserves performance and eases implementation but also reduces the memory and computation required for reconstruction and projection operations during optimization.
>
> > **W4**: The convergence theory is largely boilerplate, relying on standard FL proofs without any MAPO-specific insight
>
> As the reviewer correctly observed, MAPO maintains the same convergence rate as established global projection schemes; this is intentional, as our goal is to demonstrate that MAPO preserves standard convergence guarantees **while providing a simpler, more efficient, model-agnostic projection framework** suitable for communication-efficient FL.
> As the notation for number layers or their dimensions is entirely outside our formulation, it demonstrates the model-agnostic and architecture-independent nature of MAPO convergence.

---

> ### Author Response · Authors · 2025-11-21
>
> ---
>
> > **W2**:  The comparison set is largely incomplete: several recent or highly relevant works are missing, such as FedEx-LoRA [1], SparseFL [2], FedRPCA [3], PRISM [4], Heterogeneous LoRA for Federated Fine-tuning [5], etc. These should have been discussed and compared empirically.
>
> Thank you for introducing recent works, we have studied them in-depth and we believe the literature you are offering addresses a different problem formulation and they are directly comparable to MAPO, but more closely resemble the advancement in parameter-factorization, PEFT, and LoRA techniques in FL, and not low-rank gradient optimization in FL.
>
> While these works can bring a level of communication efficiency in FL, they are not, in essence, a low-rank compression or optimization technique comparable to our work.
>
> Below, we detail this fact.
>
> **FedEX-LoRA / FedRPCA**: These methods address the LoRA averaging problem in FL. Our process does not suffer from the low-rank averaging problem, since the A matrices are fixed; thus, averaging low-rank B matrices is identical to full-rank averaging.
> Additionally, they are focused on LoRA architecture for parameter factorization and PEFT use cases. Our work methodology and objectives are entirely different, focusing on optimization via gradient factorization and communication efficiency.
>
> **HETLORA**: Similar to the previous paper, the methodology and objective are entirely different from our work. They address the problem of aggregating clients' LoRA adaptors, given their heterogeneous ranks. In our work, we do not face such problems, and the methodology is unrelated to our study.
>
> **SparseFL**: This method applies a parallel approach to low-rank approximation, in which they restrict the optimization to a smaller number of parameters with masking, instead of projection.
> This means one can stack these methods on top of each other, and we can consider a sparse learning of the projection matrix of MAPO to enhance the communication efficiency further.
> We focus our main baselines on methods that rely on low-rank approximation and show that our approach of using low-rank factorization can outperform other styles of low-rank optimization or compression.
>
> The main contribution of SparseFL is the improvement over previous sparse learning algorithms by introducing a mask consciousness, and the final performance only improved the communication 10 to 20 times on MNIST, FMNIST, CIFAR10, and CIFAR-100. MAPO achieves superior results, with improvements of 30-100 times for those datasets.
>
> **PRISM**: This work addresses completely different problems of FL about resource construction (memory and computation) at clients. We provide a solution where each client trains a part of the model when full model training is impossible for the clients. On Page 8, Remark 3.3, they explicitly acknowledge that "PriSM is not a low-rank compression method."

---

> ### Author Response · Authors · 2025-11-21
>
> > **W3**:  There is no rigorous analysis of computational cost.
>
> We thank the reviewer for raising this concern. As clarified in our revised Appendix K, the additional computation introduced by MAPO is indeed present. Still, it remains marginal in practice and is significantly lower than that of layer-based low-rank methods.
>
> MAPO performs reconstruction and projection using a single matrix multiplication. The full gradient vector is reshaped into $W \in \mathbb{R}^{k \times \lceil d/k \rceil}$ and factorized by a projection vector $A \in \mathbb{R}^{1 \times \lceil d/k \rceil}$ and a reconstruction vector $B \in \mathbb{R}^{k \times 1}$. Both projection and reconstruction have cost $O(\lceil d/k \rceil \times k \times 1) = O(d)$. Since each operation requires one multiplication and one addition per element, the total FLOPs per optimization step is $4d$. Relative to the cost of a standard forward plus backward pass, approximately $6bd$, MAPO's overhead becomes $\frac{4d}{6bd} = \frac{2}{3b}$, which is independent of $d$ and remains near $2.1$% for $b=32$ and decreases further for larger $b$.
>
> In EvoFed, the full model update is partitioned into $p$ parts and each factorized separately with population rank $q$. We can write it as $W \in \mathbb{R}^{p \times \lceil d/p \rceil}$ factorized by a projection vector $A \in \mathbb{R}^{q \times \lceil d/p \rceil}$ and a reconstruction vector $B \in \mathbb{R}^{p \times q}$.
> Both projection and reconstruction have cost $O(\lceil d/p \rceil \times q \times p) = O(dq),$
> which is applied once per round rather than once per batch. If each client performs \(e\) local epochs per round, the relative overhead becomes $\frac{4dq}{6bed} = \frac{2q}{3be}$. Which is negligible for realistic values of the hyperparameter. For instance, if $b=32, e=200, p=\frac{d}{10}$, and $q=10$ the overhead is around $0.1$%. The overhead decreases for larger $b$ and $e$ while increasing with the number of population $q$, and is independent of the number of partitions $p$.
>
> For FedLoRU, the additional computation is performed layer-wise. If the $i$th layer has shape $W_i \in \mathbb{R}^{d_i^1 \times d_i^2}$ and rank $q$, the layer update requires $O(d_i^1 \times q \times d_i^2)$ FLOPs. Summed across layers, this becomes $O(qd)$, yielding an additional $4qd$ FLOPs per batch. Relative to $6bd$, the overhead is $\frac{4qd}{6bd} = \frac{2q}{3b}$, which grows linearly with $q$. For example, $q=4$ and $b=32$ produce about $8.3$% overhead. This highlights that MAPO provides a factor of $q$ reduction in computation compared to FedLoRU.
>
> Empirical wall-clock measurements using a single NVIDIA RTX 3090 confirm the theoretical analysis. MAPO's overhead remains at or below two percent across all datasets. EvoFed remains within fractions of a percent, while FedLoRU incurs significantly higher overhead, particularly for larger models such as TinyImageNet and GLUE. These measurements validate that MAPO's computation overhead is negligible in practical training settings.
>
> |Dataset|Batch size|FedAvg|MAPO|Over Head|EvoFed|Over Head|FedLoRU|Over Head|
> |-|-|-|-|-|-|-|-|-|
> |MNIST|32|5.277|5.347|1.33%|5.28|0.06%|5.439|3.07%|
> |FMNIST|32|5.277|5.356|1.50%|5.283|0.11%|5.468|3.62%|
> |C10|32|47.393|48.173|1.64%|47.435|0.09%|50.143|5.80%|
> |C100|32|67.705|68.75|1.54%|67.724|0.03%|71.139|5.07%|
> |Tiny|32|117.965|119.409|1.22%|118.044|0.06%|130.517|10.64%|
> |Shak|32|0.634|0.642|1.26%|0.635|0.16%|0.656|3.47%|
> |Sent|32|0.304|0.308|1.40%|0.305|0.33%|0.314|3.29%|
> |SVHN|32|32.478|32.961|1.49%|32.49|0.04%|34.061|4.87%|
> |GLUE|128|127.835|128.924|0.85%|128.056|0.17%|146.989|14.98%|
>
> These new results are included in Appendix K of the revised manuscript.
>
> Memory overhead
> ---
>
> MAPO introduces two vectors, $A \in \mathbb{R}^{1 \times \lceil d/k \rceil}$ and $B \in \mathbb{R}^{k \times 1}$, whose parameter plus activation memory is $2(\frac{d}{k} + k)$. Since the base model requires approximately $2d$ units of memory, the relative memory overhead of MAPO is $\frac{1}{k} + \frac{k}{d}$. For a typical choice such as $k=d/100$, this evaluates to slightly above $1$%.
>
> In EvoFed, the matrix $W \in \mathbb{R}^{p \times \lceil d/p \rceil}$ is factorized by a projection vector $A \in \mathbb{R}^{q \times \lceil d/p \rceil}$ and a reconstruction vector $B \in \mathbb{R}^{p \times q}$. EvoFed does not store activations, therefore the total memory is $q(\frac{d}{p}+p)$ and the relative memory overhead will be $\frac{q}{2}(\frac{1}{p}+\frac{p}{d})$. Given the practical values of $q=10, p=\frac{d}{10}$ we get $5(\frac{10}{d} + \frac{1}{10}) \approx 50$%.
>
> FedLoRU introduces per-layer low-rank matrices with total parameter count $qd_i^1 + qd_i^2$. Under the simplifying assumption $d_i^1=d_i^2=d_i$ and a matched communication budget with MAPO, where $k = q\sum_i d_i$, the total memory for parameters and activations becomes $4k$.
>
> The relative overhead is then $\frac{4k}{2d} = \frac{2k}{d}$, which corresponds to $2$% when $k = d/100$.

---

> > ### Author Response · Authors · 2025-11-28
> >
> > Thank you very much for taking the time to review our work and provide your feedback. We hope our answers are satisfactory. If you have any further questions or concerns, we would be more than happy to respond at any time.

---

### Official Review · Reviewer_zkJ3 · 2025-10-31

**Soundness:** 3
**Presentation:** 3
**Contribution:** 3
**Rating:** 4
**Confidence:** 4

**Summary:**

The paper introduces Model-Agnostic Projection Optimization (MAPO), a novel method for communication-efficient federated learning (FL). The core idea is to circumvent the limitations of traditional layer-wise low-rank decomposition. Instead of factorizing each layer's gradient, MAPO reshapes the entire model's gradient vector into a single matrix. This matrix is then factorized into a small, trainable projection vector ($B$) and a larger, non-trainable reconstruction matrix ($A$). The matrix $A$ is not transmitted; instead, it is regenerated on all clients and the server in each round using a shared random seed. The authors claim this approach is model-agnostic, offers more flexible control over the communication budget, and improves performance by exploring new random subspaces in each round. The paper provides a theoretical convergence analysis and extensive empirical results across various datasets and models to support its claims.

**Strengths:**

1. The central idea of reshaping the entire model gradient into a single matrix and applying a rank-1-like factorization is a conceptually simple yet novel departure from the prevalent layer-wise decomposition methods in FL (like LoRA variants for gradients). The use of a shared seed to synchronize a dynamically changing projection basis ($A$) without communication cost is a clever and effective mechanism.

2. The empirical results, if taken at face value, are highly significant. The method demonstrates substantial communication savings, reportedly achieving performance close to uncompressed FedAvg with orders of magnitude less communication (e.g., 98.9% of FedAvg's accuracy on CIFAR-10 with only 1.2% of the communication cost). This could represent a major step forward for practical, large-scale FL deployments.

3. The paper is well-written and the core idea is presented clearly, particularly with the aid of Figure 1 and Algorithm 1. The authors have conducted a comprehensive empirical evaluation, spanning a wide variety of datasets (vision, NLP), model architectures (CNN, Transformer, RoBERTa), and data distributions (non-IID). The inclusion of numerous appendices covering fine-tuning, complexity analysis, and ablation studies indicates a thorough investigation. The ablation study in Figure 7, which highlights the benefit of a "Fresh A" over a "Frozen A," provides strong support for the subspace exploration argument.

**Weaknesses:**

1. The "model-agnostic" reshaping, while simple, is heuristically motivated and lacks a strong theoretical foundation. It concatenates gradients from disparate layers (e.g., convolutional filters, attention heads, normalization parameters) which have fundamentally different structures, scales, and sensitivities. The paper provides no theoretical justification or empirical analysis as to why this aggressive homogenization is a principled approach. It is plausible that this could harm optimization by mixing unrelated gradient information, and the method's success might be more attributable to the sheer reduction in variance from projecting onto a random subspace rather than any inherent benefit of the reshaping itself.

2. The paper focuses heavily on communication efficiency but downplays the computational overhead introduced on the client side. During local training, each forward and backward pass requires reconstructing the model update from the projection, i.e., computing $\Delta W' = \text{vec}(B A^T)[0:d]$. This involves a matrix multiplication and reshaping operation within the local training loop for every batch. While Appendix K provides a theoretical complexity analysis, it is abstract. A practical wall-clock time comparison against baselines is conspicuously absent. For large models where $d$ is massive, even if $k$ is small, the dimension of $A$ ($\approx d/k$) can be very large, making the BA product a potential bottleneck that could negate the benefits in scenarios where computation, not just communication, is constrained.

3.The theoretical claims, particularly Proposition 3.6 regarding a memory reduction of $k^2$, are potentially misleading. The analysis in Appendix K correctly shows that the memory to store the factorization components ($A$ and $B$) is reduced. However, it ignores the dominant memory cost of the full-rank model parameters ($W$) and activations, which MAPO must maintain on the client. The method is communication-efficient, not parameter-efficient (unlike LoRA for fine-tuning). This crucial distinction is not made sufficiently clear in the main text, and the claim of significant memory reduction could be misinterpreted by readers.

**Questions:**

1. Could you provide wall-clock time comparisons for local client training (per round) between MAPO and the main baselines (FedAvg, FedLoRU, EvoFed)? This would clarify the practical computational trade-offs of your method.

2. The core "reshaping" mechanism treats all parameters equally. Have you investigated the effect of this homogenization? For instance, does normalizing gradients from different layers before concatenation affect performance? Is there any intuition for why mixing gradients from a shallow convolutional layer and a deep classification layer is beneficial?

3. Could you clarify the claim of $k^2$ memory reduction in Proposition 3.6 in the context of the overall client memory footprint? Given that the full model $W$ of size $d$ must be stored, what is the practical memory saving as a percentage of the total memory required to train the model on a client?

4. How sensitive is MAPO's performance to the initial random seed $r_t$? Does the choice of a "lucky" or "unlucky" random projection matrix $A_t$ in a given round lead to high variance in the training trajectory?

---

> ### Author Response · Authors · 2025-11-21
>
> Thank you for your thoughtful and constructive review. We appreciate your recognition of the **novelty of reshaping**, the **effectiveness of the synchronized random basis**, the **significance of our communication savings**, and the **clarity** and **breadth of our empirical evaluation**.
> We value your assessment, and we believe our responses and added experiments strengthen the work.
>
> ---
> > **W1:** The paper provides no theoretical justification or empirical analysis as to why this aggressive homogenization is a principled approach. It is plausible that this could harm optimization by mixing unrelated gradient information
>
> We appreciate the reviewer’s concern. In Section 3, Definitions 3.2–3.3 and Propositions 3.4–3.6 formally show that, given a **1) fixed random Gaussian reconstruction matrix** and under a **2) fixed communication budget**, the **expected reconstruction error rate** remains the _same_ whether projection is done layer-wise, on the whole model, or on the reshaped full-model gradient.
>
> Intuitively, because a Gaussian random subspace is isotropic and uncorrelated with the structure of any particular layer, factorization does not benefit from exploiting layer-specific geometry. Therefore, mixing gradients from different layers does not introduce additional bias relative to standard layer-wise random projection.
>
> This is consistent with prior works that use random subspaces (e.g., EvoFed, random sketching) and contrasts with approaches that rely on _gradient subspace_ structures rather than Gaussian random subspaces, where layer-wise projection is indeed beneficial.
>
> Propositions do **not** claim an improvement over layer-wise factorization, instead they  show that reshaping is **theoretically equivalent** to prior projection methods in expected reconstruction error while offering a reduction in reconstruction-matrix memory cost and enabling a simple, model-agnostic implementation.
>
> > **Q2:**  The core "reshaping" mechanism treats all parameters equally. Have you investigated the effect of this homogenization? For instance, does normalizing gradients from different layers before concatenation affect performance? Is there any intuition for why mixing gradients from a shallow convolutional layer and a deep classification layer is beneficial?
>
> We appreciate the reviewer’s insightful question.
> Firstly, as discussed in Section 2 (“Sketching Limitations”), there is a key distinction between _post-hoc projection_ (e.g., gradient sketching after local training, EvoFed) and _structured updates_, where the low-rank constraint is applied **inside** the optimization loop.
>
> In pure sketching, treating layers uniformly can be problematic, as the reviewer mentioned, because the projection is applied after the fact, without reference to the loss landscape.
> However, in MAPO, the projection vector $B_t$ is updated via many small steps of _gradient descent_, and therefore the learning dynamics automatically emphasize parameters and layers that have higher influence on the loss at each batch update.
> While the reconstruction matrix $A_t$ is fixed by the seed, the update direction $B_t$ is explicitly optimized to minimize the loss objective, meaning that the model does **not** rely on uniform contributions and representation across layers.
>
> In other words, while post-hoc projection aims to approximate and reconstruct the full model update (minimizing the error distance), our method's objective is to find a subspace solution that minimizes the loss function, even if the final update is not the closest to a full-rank update.
>
> As $k \ll d$, the random subspace during optimization is almost orthogonal to the full model update, our final model update shows almost zero cosine similarity with the full model gradient, suggesting that the MAPO solution is not an approximation of the full-rank solution, instead an alternative solution that can be generated in a random subspace.
>
>
> Secondly, layer-wise low-rank methods allocate a fixed rank (and therefore a fixed communication budget) per layer, regardless of the amount of useful signal in that layer at a given batch. This static allocation is often suboptimal, as each layer update contributes differently for each batch.
> In contrast, MAPO performs factorization at the full-model level. The projection vector $B_t$ naturally concentrates on the directions (across _any_ layer) that most reduce the loss at each step. This gives MAPO the ability to dynamically allocate additional capacity to layers that carry stronger gradient signals in that batch.

---

> ### Author Response · Authors · 2025-11-21
>
> > **W2**: The paper focuses heavily on communication efficiency but downplays the computational overhead introduced on the client side.
> > **Q1**: 1.  Could you provide wall-clock time comparisons for local client training (per round) between MAPO and the main baselines (FedAvg, FedLoRU, EvoFed)? This would clarify the practical computational trade-offs of your method.
>
> We thank the reviewer for raising this concern.
> As we explained in the revised Appendix K, the additional computation introduced by MAPO is indeed present, but it remains marginal in practice and is even lower than that of layer-wise low-rank methods.
>
> ### Computational Overhead
>
> MAPO performs reconstruction and projection using a single matrix multiplication. The full gradient vector is reshaped into $W \in \mathbb{R}^{k \times \lceil d/k \rceil}$ and factorized by a projection vector $A \in \mathbb{R}^{1 \times \lceil d/k \rceil}$ and a reconstruction vector $B \in \mathbb{R}^{k \times 1}$. Both projection and reconstruction have cost $O(\lceil d/k \rceil \times k \times 1) = O(d)$. Since each operation requires one multiplication and one addition per element, the total FLOPs per optimization step is $4d$. Relative to the cost of a standard forward plus backward pass, approximately $6bd$, MAPO's overhead becomes $\frac{4d}{6bd} = \frac{2}{3b}$, which is independent of $d$ and remains near $2.1\$% for $b=32$ and decreases further for larger $b$.
>
> In EvoFed, the full model update is partitioned into $p$ parts and each factorized separately with population rank $q$. We can write it as $W \in \mathbb{R}^{p \times \lceil d/p \rceil}$ factorized by a projection vector $A \in \mathbb{R}^{q \times \lceil d/p \rceil}$ and a reconstruction vector $B \in \mathbb{R}^{p \times q}$.
> Both projection and reconstruction have cost $O(\lceil d/p \rceil \times q \times p) = O(dq),$
> which is applied once per round rather than once per batch. If each client performs \(e\) local epochs per round, the relative overhead becomes $\frac{4dq}{6bed} = \frac{2q}{3be}$. Which is negligible for realistic values hyperparameter. For instance, if $b=32, e=200, p=\frac{d}{10}$, and $q=10$ the overhead is around $0.1\$%. The overhead decreases with larger $b$ and $e$, increases with the population size $q$, and is independent of the number of partitions $p$.
>
> For FedLoRU, the additional computation is performed layer-wise. If the $i$th layer has shape $W_i \in \mathbb{R}^{d_i^1 \times d_i^2}$ and rank $q$, the layer update requires $O(d_i^1 \times q \times d_i^2)$ FLOPs. Summed across layers, this becomes $O(qd)$, yielding an additional $4qd$ FLOPs per batch. Relative to $6bd$, the overhead is $\frac{4qd}{6bd} = \frac{2q}{3b}$, which grows linearly with $q$. For example, $q=4$ and $b=32$ produce about $8.3\$% overhead. This highlights that MAPO provides a factor of $q$ reduction in computation compared to FedLoRU.
>
> Empirical wall-clock measurements using a single NVIDIA RTX 3090 confirm the theoretical analysis. MAPO's overhead remains at or below two percent across all datasets. EvoFed remains within fractions of a percent, while FedLoRU incurs significantly higher overhead, particularly for larger models such as TinyImageNet and GLUE. These measurements validate that MAPO's computation overhead is negligible in practical training settings.
>
> |Dataset|Batch size|FedAvg|MAPO|Over Head|EvoFed|Over Head|FedLoRU|Over Head|
> |-|-|-|-|-|-|-|-|-|
> |MNIST|32|5.277|5.347|1.33%|5.28|0.06%|5.439|3.07%|
> |FMNIST|32|5.277|5.356|1.50%|5.283|0.11%|5.468|3.62%|
> |C10|32|47.393|48.173|1.64%|47.435|0.09%|50.143|5.80%|
> |C100|32|67.705|68.75|1.54%|67.724|0.03%|71.139|5.07%|
> |Tiny|32|117.965|119.409|1.22%|118.044|0.06%|130.517|10.64%|
> |Shak|32|0.634|0.642|1.26%|0.635|0.16%|0.656|3.47%|
> |Sent|32|0.304|0.308|1.40%|0.305|0.33%|0.314|3.29%|
> |SVHN|32|32.478|32.961|1.49%|32.49|0.04%|34.061|4.87%|
> |GLUE|128|127.835|128.924|0.85%|128.056|0.17%|146.989|14.98%|
>
> These new results are included in Appendix K of the revised manuscript.
>
> ---
> > **W3**:  The theoretical claims, particularly Proposition 3.6 regarding a memory reduction of, are potentially misleading.
>
> We appreciate the reviewer highlighting the potential for misinterpreting Proposition 3.6. We fully agree with your assessment: **MAPO is communication-efficient, not parameter-efficient**, and it does _not_ reduce the dominant memory cost of storing the full model parameters or activations during local training. As noted in **Table 1**, **Section 2.2**, and **Table 4 in Appendix B**, MAPO maintains a similar parameter footprint to FedAvg, unlike LoRA-style fine-tuning methods that explicitly aim for parameter efficiency.
> We updated Proposition 3.6 to explicitly state that the $k^2$ reduction in memory of the reconstruction matrix $A$.
> Additionally, we revised Appendix K to incorporate the full model parameters and activation memory into the analysis for completeness, clearly showing the overhead introduced by each method.

---

> ### Author Response · Authors · 2025-11-21
>
> > **Q3**:  Could you clarify the claim of memory reduction in Proposition 3.6 in the context of the overall client memory footprint? Given that the whole model of size must be stored, what is the practical memory saving as a percentage of the total memory required to train the model on a client?
>
> We revised Appendix K to include the details of practical memory overhead, and we provide a brief explanation below.
>
> MAPO introduces two vectors, $A \in \mathbb{R}^{1 \times \lceil d/k \rceil}$ and $B \in \mathbb{R}^{k \times 1}$, whose parameter plus activation memory is $2(\frac{d}{k} + k)$. Since the base model requires approximately $2d$ units of memory, the relative memory overhead of MAPO is $\frac{1}{k} + \frac{k}{d}$. For a typical choice such as $k=d/100$, this evaluates to slightly above $1$%.
>
> In **EvoFed** the matrix $W \in \mathbb{R}^{p \times \lceil d/p \rceil}$ factorized by a projection vector $A \in \mathbb{R}^{q \times \lceil d/p \rceil}$ and a reconstruction vector $B \in \mathbb{R}^{p \times q}$. EvoFed does not store activations, therefore the total memory is $q(\frac{d}{p}+p)$ and the relative memory overhead will be $\frac{q}{2}(\frac{1}{p}+\frac{p}{d})$ . Given the practical values of $q=10, p=\frac{d}{10}$ we get $5(\frac{10}{d} + \frac{1}{10}) \approx 50$%.
>
> In contrast, **FedLoRU** introduces per-layer low-rank matrices with total parameter count $qd_i^1 + qd_i^2$. Under the simplifying assumption $d_i^1=d_i^2=d_i$ and a matched communication budget with MAPO, where $k = q\sum_i d_i$, the total memory for parameters and activations becomes $4k$.
>
> The relative overhead is then $\frac{4k}{2d} = \frac{2k}{d}$, which corresponds to $2$% when $k = d/100$.
>
> These analyses and measurements confirm that MAPO’s computation and memory overheads are minimal and do not diminish its communication benefits.
> These new results are included in Appendix K of the revised manuscript
>
> > **Q4**:  How sensitive is MAPO's performance to the initial random seed? Does the choice of a "lucky" or "unlucky" random projection matrix in a given round lead to high variance in the training trajectory?
>
> Across all of our experiments, MAPO exhibits **very low sensitivity** to the choice of random seed, as variances are shown in Table 3. Because MAPO regenerates a new projection basis $A_t$ _every round_, the effects of individual “lucky” or “unlucky” random subspaces are averaged out over training. In our repeated trials with different seeds, we consistently observe **only minor variance**, typically within the natural stochasticity of FedAvg itself. This matches our intuition: since MAPO explores many random subspaces across rounds, the training trajectory is not dominated by any single projection matrix.
>
> We also conducted an expanded study on **Frozen-A**, in which the random projection basis is fixed throughout the training run. This condition isolates the effect of seed choice directly. Here, we do observe a seed-dependent pattern:
>
> -   When the subspace is **large** (e.g., $k \ge d/2$), a Frozen-$A$ behaves similarly to full-rank training. We can see the effects of "lucky" and "unlucky" seeds: some fraction of seeds underperform significantly, while others outperform Fresh-A.
>
> -   When the subspace is realistic and **small**, Frozen-A consistently converges to a **suboptimal local minimum**, and the variance is noticeably larger than Fresh-A.
>
> Below, we provide illustrative seed-sweep results (Fresh-A vs Frozen-A) that reflect this pattern for 10 seeds on the FMNIST dataset:
>
> |Method|k/d|Std|Min|Max|
> |-|-|-|-|-|
> |Fresh-A|1/2|0.0021|0.8838|0.8900|
> |Frozen-A|1/2|0.0898|0.6720|0.8992|
> |Fresh-A|1/100|0.0035|0.8733|0.8836|
> |Frozen-A|1/100|0.0115|0.7389|0.7711|
>
> These new findings are in Appendix G.1.

---

> > ### Author Response · Authors · 2025-11-28
> >
> > Thank you very much for taking the time to review our work and provide your feedback. We hope our answers are satisfactory. If you have any further questions or concerns, we would be more than happy to respond at any time.

---

### Meta-Review · Area_Chair_DFp4 · 2026-01-06

**Summary:**

This paper proposes a method called Model-Agnostic Projection Optimization (MAPO) that reshapes and factorizes the full model gradient into a fixed reconstruction matrix and a trainable projection vector to reduce the communication cost of FL.

**Reviewer Concerns:**

- The novelty is limited, as MAPO adopts well known techniques such as global random projection and seed synchronization.
- The convergence analysis is standard and the paper does not provide any theoretical justification or insight on the benefit of MAPO.
- The paper does not characterize additional computational or storage costs of MAPO.

During the rebuttal, the authors provide additional computational and memory cost analysis. However, the concerns on the technical novelty and limited theoretical contribution still remain.

**Reviewer Scores:**

The original scores were 4/2/4/6. The reviewers would likely maintain their scores.

---

### Decision · Program_Chairs · 2026-01-26

Reject